# CaPo: Cooperative Plan Optimization for Efficient Embodied Multi-Agent Cooperation

**Jie Liu**[1]   **Pan Zhou**[2]*   **Yingjun Du**[1]   **Ah-Hwee Tan**[2]   **Cees G.M. Snoek**[1]
**Jan-Jakob Sonke**[3]   **Efstratios Gavves**[1,4]
[1]University of Amsterdam, The Netherlands   [2]Singapore Management University, Singapore
[3]The Netherlands Cancer Institute , The Netherlands   [4]Archimedes/Athena RC, Greece
`j.liu5@uva.nl`  `panzhou3@gmail.com`  `E.Gavves@uva.nl`

## Abstract

In this work, we address the cooperation problem among large language model (LLM) based embodied agents, where agents must cooperate to achieve a common goal. Previous methods often execute actions extemporaneously and incoherently, without long-term strategic and cooperative planning, leading to redundant steps, failures, and even serious repercussions in complex tasks like search-and-rescue missions where discussion and cooperative plan are crucial. To solve this issue, we propose Cooperative Plan Optimization (CaPo) to enhance the cooperation efficiency of LLM-based embodied agents. Inspired by human cooperation schemes, CaPo improves cooperation efficiency with two phases: 1) meta-plan generation, and 2) progress-adaptive meta-plan and execution. In the first phase, all agents analyze the task, discuss, and cooperatively create a meta-plan that decomposes the task into subtasks with detailed steps, ensuring a long-term strategic and coherent plan for efficient coordination. In the second phase, agents execute tasks according to the meta-plan and dynamically adjust it based on their latest progress (e.g., discovering a target object) through multi-turn discussions. This progress-based adaptation eliminates redundant actions, improving the overall cooperation efficiency of agents. Experimental results on the ThreeDworld Multi-Agent Transport and Communicative Watch-And-Help tasks demonstrate CaPo's much higher task completion rate and efficiency compared with state-of-the-arts. The code is released at https://github.com/jliu4ai/CaPo.

## 1 Introduction

Large Language Models (LLMs) have demonstrated remarkable capabilities in understanding and generating human language, complex reasoning, and planning, achieving impressive advancement (OpenAI, 2024; Touvron et al., 2023). These advancements empower LLM-based embodied agents to autonomously make plans (Li et al., 2023a; Padmakumar et al., 2022; Zhu et al., 2023; Wang et al., 2023a; Wu et al., 2023b; Huang et al., 2022b) and perform reasoning (Du et al., 2023; Hao et al., 2023; Zhou et al., 2024; Huang et al., 2022a) by using human language to assist people in daily activities, such as housework and daily chores. The next milestone for agents is to cooperate with others to achieve joint tasks. This is crucial not only for efficiently performing simple tasks but also for tackling complex ones that cannot be completed in isolation due to their inherent complexity or the dynamic nature of the environment (Zhang et al., 2023b; Guo et al., 2024; Mandi et al., 2023).

Notably, the cooperation among LLM-based embodied agents is rarely investigated despite being highly desired. Conventional works often focus on adopting reinforcement learning (RL) (Jiang & Lu, 2018; Liu et al., 2021; Wang et al., 2021) to explore the dynamics of cooperative behavior among non-LLM-based agents. In spite of their promising performance in certain scenarios, RL-based cooperation methods exhibit limited adaptability across different tasks (Dittadi et al., 2021; Cobbe et al., 2019), since they are often not trained on large-scale data and lack sufficient generalization ability. To solve this issue, in this work, we are particularly interested in the problem of "how to

---

*Corresponding author

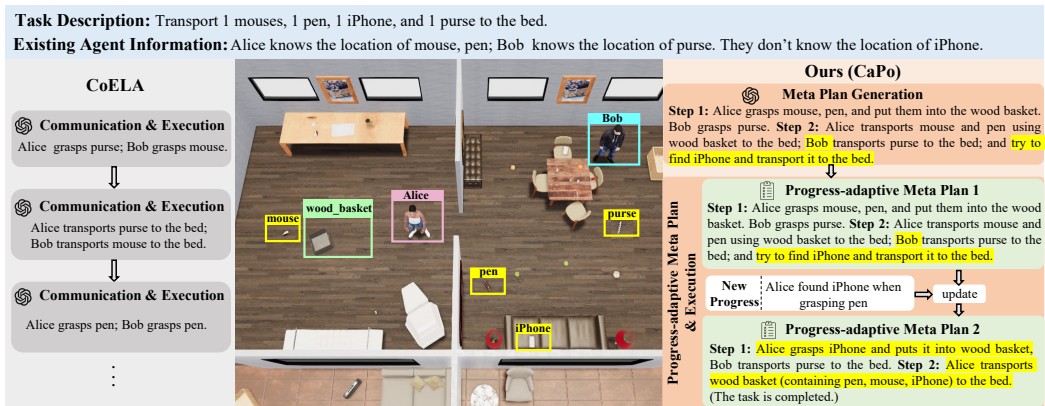

Figure 1: **Examples of task accomplishment of CoELA (Zhang et al., 2023b) and our CaPo**. In CoELA, after each action execution, Alice and Bob communicate to decide next action which is a greedy single-step plan and suboptimal. For example, they do not use wood basket which can contain multiple objects, and both extemporaneously move a single item to the target bed without a long-term and collaborative plan. In contrast, in CaPo, Alice and Bob first discuss to make a long-term meta-plan for strategical cooperation in which Alice is arranged to move several target items into a wood basket, and Bob moves the remaining target items and also searches the unknown objects. Then during execution phase, both follow the meta-plan to accomplish task, and dynamically adapt the meta-plan to latest task progress, ensuring its effectiveness and efficiency in coordinating agents.

develop an effective collaboration framework for LLM-based agents", since LLMs have revealed strong reasoning and planning capability as good agents' brain across different tasks.

Among the limited related works, CoELA (Zhang et al., 2023b) proposes an LLM-based multi-agent cooperation framework in which after each action execution, agents communicate to devise a single-step plan for the next action. Despite its significant advancements, CoELA's short-term, single-step planning, which *lacks consideration for long-term strategic collaboration*, often results in extemporaneous and incoherent actions among agents, leading to several potential issues. ①  Without a long-term coherent collaboration plan, agents are prone to executing numerous redundant actions, resulting in increased costs. This issue is particularly critical in the physical world, where agents' movements are inherently challenging and resource-intensive. For instance, as shown in Fig. 1, for the object transport task, agent Alice and Bob do not use the wood basket which can contain several objects, and extemporaneously move their nearest target objects one by one, leading to inferior efficiency. ② Complex tasks are challenging to accomplish without thorough discussion and long-term collaboration, particularly in embodied environments where agents operate with only partial observations. ③ Without a long-term cooperative plan, agents' uncoordinated actions can lead to severe consequences. For example, in search-and-rescue missions, poor coordination may endanger lives due to the complexity of such operations.

**Contributions.** This work introduces *Cooperative Plan Optimization (CaPo)*, a novel framework designed to enhance the cooperation efficiency of LLM-based embodied agents by leveraging their advanced reasoning and planning capabilities. Inspired by human cooperation schemes (Tuomela, 1998; Thürmer et al., 2017), CaPo enables agents to collaboratively create and refine long-term, strategic, and coherent meta-plans through multi-turn discussions. These meta-plans serve as actionable guides that facilitate efficient task coordination and completion.

Specifically, CaPo addresses key challenges in multi-agent cooperation through two main phases. The first phase is **Meta-Plan Generation**, where agents collaboratively analyze the task and exchange information to generate a meta-plan that decomposes the overall task into detailed subtasks, including agent assignments and execution steps. A designated agent initiates the meta-plan, while others evaluate and refine it through iterative feedback, ensuring alignment and optimal planning. The discussion process continues until consensus is reached or the communication budget is exhausted. This phase ensures thorough deliberation and creates a coherent roadmap for task execution. For example, in the object transport task, agents Alice and Bob are strategically assigned complementary subtasks to optimize their coordination (see Fig. 1).

The second phase is **Progress-Adaptive Meta-Plan and Execution**, where agents dynamically adapt the meta-plan based on real-time progress and discoveries during task execution. For instance, if Alice identifies the location of an object that was initially unknown (e.g., an "iPhone"), the agents collaboratively update the meta-plan to incorporate this new information, ensuring timely and efficient task completion. This adaptive mechanism maintains the meta-plan's effectiveness, even in dynamic environments, and prevents inefficiencies or redundant actions.

The proposed approach has been rigorously evaluated on benchmark tasks, including the *ThreeDworld Multi-Agent Transport (TDW-MAT)* and *Communicative Watch-And-Help (C-WAH)* tasks (Zhang et al., 2023b). Experimental results demonstrate that CaPo significantly improves task completion rates and cooperation efficiency compared to state-of-the-art methods. Notably, on the TDW-MAT task, CaPo achieves a completion rate improvement of 16.7% and 4.7% over the SoTA CoELA method with GPT-3.5 and GPT-4 agents, respectively.

## 2 RELATED WORK

**LLM-based Agents.** LLM-based agents (Hong et al., 2023; Wang et al., 2024; Shen et al., 2024; Liu et al., 2023a) are designed to autonomously perceive environments, execute actions, and accumulate knowledge, with rich real world knowledge and complex reasoning capability inherited from LLMs. Notable agents like AutoGPT (Richards & et al, 2021), BabyAGI (Nakajima, 2023), and AgentGPT (Reworkd, 2023) showcase remarkable proficiency in decision-making and complex reasoning. In the embodied environment, LLM-based agents have shown superior capacity in strategic planning (Li et al., 2023a; Padmakumar et al., 2022; Wu et al., 2023b; Huang et al., 2022b). Specifically, LLM-planner (Song et al., 2023) harness LLMs to do few-shot planning for embodied agents. PET (Wu et al., 2023a) translates a task description with LLMs into a list of high-level sub-tasks. TaPA wu2023embodied enables the agent to generate executable plans by aligning LLMs with visual perception models. Another research line focuses on harnessing LLMs's reasoning capabilities in embodied tasks (Zhou et al., 2024; Huang et al., 2022a). ELLM (Du et al., 2023) utilizes LLMs to set pretraining goals in RL, guiding agents towards the goal without human involvement.

**Multi-Agent Cooperation.** Multi-agent cooperation and communication have been studied for decades to improve communication efficiency (Jiang & Lu, 2018; Li et al., 2023b) and planning (Torreno et al., 2017; Zhang et al., 2023a). Within the domain of embodied intelligence, ProAgent (Zhang et al., 2023a) harnesses LLMs to develop proactive agents that dynamically adjust their behavior to foster better cooperation with teammates. RoCo (Mandi et al., 2023) introduce a multi-robot collaboration framework that employs LLMs for both high-level communication and low-level path planning. (Guo et al., 2024) proposed a prompt-based organizational framework for LLM agents to reduce communication costs and boost team efficiency. CoELA (Zhang et al., 2023b) enables agents to plan, communicate, and collaborate effectively, but its plan is one-step plan and is short-term. Despite these advancements, these methods focus on short-term planning and do not involve sufficient agent discussion, while ours seeks to a long-term strategical and coherent plan via agent's thoughtful discussions for efficient multi-agent cooperation.

**Optimization with LLMs.** With the advancement of prompting techniques, LLMs have shown remarkable performance across various domains (Wei et al., 2022; Kojima et al., 2022; Wang et al., 2022; Zhou et al., 2022; Madaan et al., 2024). Their ability to understand natural language lays out a new possibility for optimization. (Yang et al., 2023) first proposed to leverage LLMs as optimizer, where the optimization task is described in natural language. OPT2I (Mañas et al., 2024) aims to enhance prompt-image consistency in text-to-image models by iteratively generating revised prompts with LLMs to maximize the consistency score. VislingInstruct (Zhu et al., 2024) proposes optimizing multi-modal instruction for multi-modal language models in a zero-shot manner. DyLAN (Liu et al., 2023b) is particularly relevant to our work. DyLAN (Liu et al., 2023b) enables agents to interact for multiple rounds in a dynamic architecture to optimize the selection of agent. In contrast, our work investigates cooperative plan optimization via multi-turn discussion between agents.

## 3 PRELIMINARIES

**Task Formulation.** We formulate the embodied multi-agent cooperation task as an decentralized partially observable Markov decision process (DEC-POMDP) (Bernstein et al., 2002; Spaan et al.,

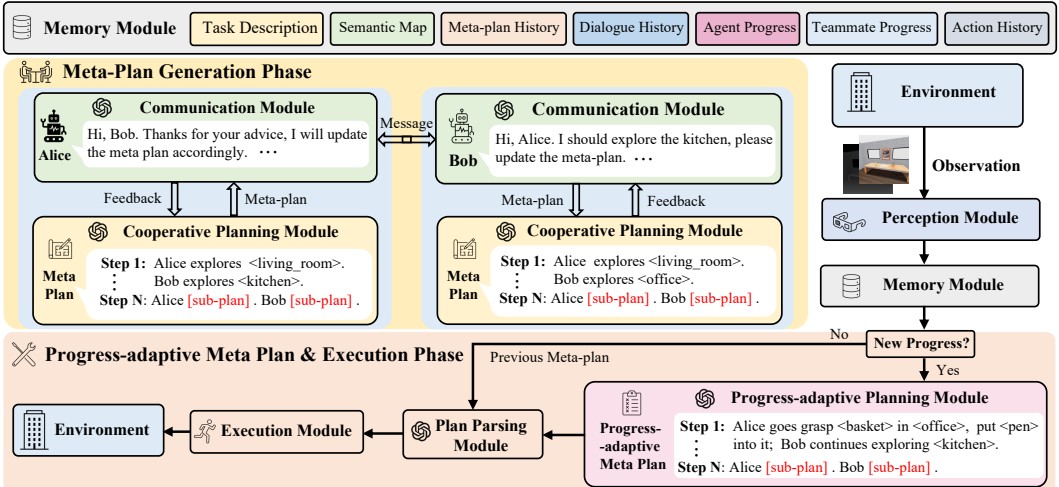

Figure 2: **Overview of the *CooperAtive Plan Optimization (CaPo)* framework for embodied multi-agent cooperation.** CaPo consists of two key phases: 1) **meta-plan Generation**: All agents collaboratively formulate a meta-plan before taking any actions through multi-turn discussions. One agent serves as meta-plan designer, responsible for creating the meta-plan, while all other agents serve as meta-plan evaluators, providing critical feedback about meta-plan. 2) **Progressive-adaptive meta-plan and Execution**: As new progress is made, agents adopt a progress-adaptive planning module to adapt the meta-plan to the latest task progress, ensuring the effectiveness of meta-plan.

2006), which is defined as $< n, \mathcal{S}, \mathcal{O}, \mathcal{A}, P, r, \gamma >$. Here, $n$ represents the number of agents; $\mathcal{S}$ is the finite state space; $\mathcal{O}$ denotes the observation space; $\mathcal{A}$ is a finite joint action space of all agents; $P : \mathcal{S} \times \mathcal{A} \times \mathcal{S} \rightarrow [0, 1]$ denotes the transition probability function; $r = \mathcal{S} \times \mathcal{A} \rightarrow \mathbb{R}$ denotes the reward function; $\gamma \in [0, 1]$ denotes the discount factor. In this framework, at time step $t \in \mathbb{N}$, each agent $i$ observes the environment's state $s_t \in \mathcal{S}$, and receives an observation set $\mathcal{O}_i$. $\mathcal{O}_i$ consists of a world observation $\mathcal{O}_i^w$, which the agent gathers through its sensors, or a communication message observation $\mathcal{O}_i^c$ from other teammate agents. Agent $i$ takes actions from its action space $\mathcal{A}_i$, which includes a finite set of world action $\mathcal{A}_i^w$, e.g., grasping a target object, or a finite set of messaging action $\mathcal{A}_i^c$. Then agents receive a shared reward $r_t = r(s_t, a_t)$, where $a_t \in \mathcal{A}$ denotes the joint actions of agents, and observe a new state $s_{t+1}$ with probability $P(s_{t+1}|s_t, a_t)$. We formulate the problem with two decentralized intelligent embodied agents working together to complete a long-horizon rearrangement task (Zhang et al., 2023b; Batra et al., 2020) in a multi-room indoor environment. During the task, agents can execute multiple kinds of actions, such as navigation, interaction, and communication by sending messages.

**CoELA Framework.** CoELA Zhang et al. (2023b) is a pioneering modular framework for embodied multi-agent cooperation, which consists of five key modules: (a) Perception, (b) Memory, (c) Communication, (d) Planning, and (e) Execution. For each agent, the (a) Perception Module gathers observations from environment, including messages from other agents and relevant scene information from the RGB-D image. The (b) Memory Module dynamically stores the shared task, dialogue history between agents, agent progress, team- mate progress, and action history, all formatted as text descriptions. (c) The Communication Module retrieves relevant information from the memory module and uses an LLM to generate messages that are sent to other agents. (d) The Planning Module driven by LLM decides which plan to take given related information retrieved from the memory module and available actions proposed regarding the current state. Finally, the (e) Execution Module convert high-level plan into primitive actions executable in the environment. CoELA exhibits great performances in the embodied multi-agent cooperation tasks, making it a strong baseline for validating the effectiveness of our model in improving multi-agent cooperation efficiency.

# 4 COOPERATIVE PLAN OPTIMIZATION

We first introduce the overall framework of CooperAtive Plan Optimization (CaPo) for LLM-based embodied agents in Sec. 4.1. We then respectively elaborate on the two key phases of CaPo, i.e., meta-plan generation and progress-adaptive meta-plan and execution, in Sec. 4.2 and Sec. 4.3.

### 4.1 OVERALL FRAMEWORK OF CAPO

CaPo addresses the challenge of enabling two centralized embodied agents to effectively cooperate in a shared environment. In this setup, each agent has partial observations and must rely on communication and coordination to accomplish complex tasks. The key objective is to achieve strategic and step-by-step collaboration, where both agents contribute to the task's completion through efficient planning and adaptive decision-making. As shown in Fig. 2, each agent is equipped with five foundational modules from CoELA: perception, memory, communication, plan-phrasing, and execution. These modules enable the agents to perceive their environment, store and retrieve relevant information, exchange messages for coordination, generate plans, and execute actions accordingly.

To complete a task cooperatively and efficiently, inspired by humans collaboration (Tuomela, 1998; Thürmer et al., 2017), CaPo first analyzes the task at hand to create a long-term meta-plan before agents take any actions. All agents participate in this plan-making process, either generating meta-plan or providing feedback. The meta-plan is then dynamically refined based on the latest agent progress to ensure its effectiveness in coordinating agents. To this end, it contains two key phases, including 1) meta-plan generation, and 2) progress-adaptive meta-plan and execution. In the meta-plan generation phase, given a task, multiple embodied agents first gather relevant information such as object locations. Then, they discuss together to create a meta-plan that decomposes the task into subtasks and consider agent situation (e.g., agent and object locations) to assign agents to different subtasks with accomplishment steps. In the progress-adaptive meta-plan and execution phase, agents dynamically align the meta-plan with their latest progress. This is achieved through multi-turn discussion triggered by clear task progress, such as discovering target objects or successfully completing subtasks. In the following, we will elaborate on these two phases in turn.

### 4.2 META-PLAN GENERATION

To generate the long-term meta-plan which coordinates all agents to accomplish tasks efficiently, CaPo introduces two key steps, including 1) meta-plan initialization where one agent initializes a meta-plan according to the task description and existing information, and 2) meta-plan evaluation and optimization where all agents evaluate the meta-plan and provide feedback to improve the plan.

**Meta-plan Initialization.** At the beginning of a task, the task description is provided to all agents, e.g, `Transport 2 apples and 3 bananas to the bed`. One agent, e.g., Alice in Fig. 2, is randomly selected as the meta-plan designer, and creates the meta-plan through a *cooperative planning module*. Note that the meta-plan here, as illustrated in Fig. 3, differs from the short-term or unorganized plans used in previous work (Zhang et al., 2023b;a; Mandi et al., 2023). Specifically, the cooperative planning module is equipped with a pre-trained LLM, and leverage the LLM to generate the meta-plan. The prompting for the LLM is organized as follows:

Prompt: `<Task Desc>` + `<Instruct Head>` \n.  LLM: `<Meta-plan>`.

Here, `<Task Desc>`, `<Instruct Head>`, and `<Meta-plan>` are three placeholders for the task description, instruction head, and generated meta-plan. The task description provides background descriptions about the task, while the instruction head introduces additional constraints into the generation of meta-plan, such as the format of meta-plan and available actions to generate a clear and executable plan. Detailed prompt design is shown in Fig. 9 of Appendix.

**Meta-plan Evaluation and Optimization.** The meta-plan generated by a single agent is often biased by that agent's partial observations, resulting in a suboptimal plan that fails to coordinate all agents effectively. To address this issue, CaPo involves all agents in a multi-turn discussion to optimize the meta-plan. Specifically, the meta-plan designer (e.g., Alice in Fig. 3) broadcasts the meta-plan to all teammate agents, while teammate agents (e.g., Bob in Fig. 3) serve as meta-plan evaluators, providing feedback about the meta-plan. Since teammate agents have different partial observations of the environment, they provide the meta-plan designer with better situational awareness to help generate a more efficient and effective meta-plan. This optimization process continues until all agents reach a consensus, i.e., the evaluator agents are satisfied with the meta-plan, or the communication budget (e.g., maximum discussion round) is exhausted. Indeed, Fig. 8 in Sec. 5.2 analyzes the convergence analysis of the meta-plan optimization process, and shows that typically agents would reach a consensus within three rounds of discussion.

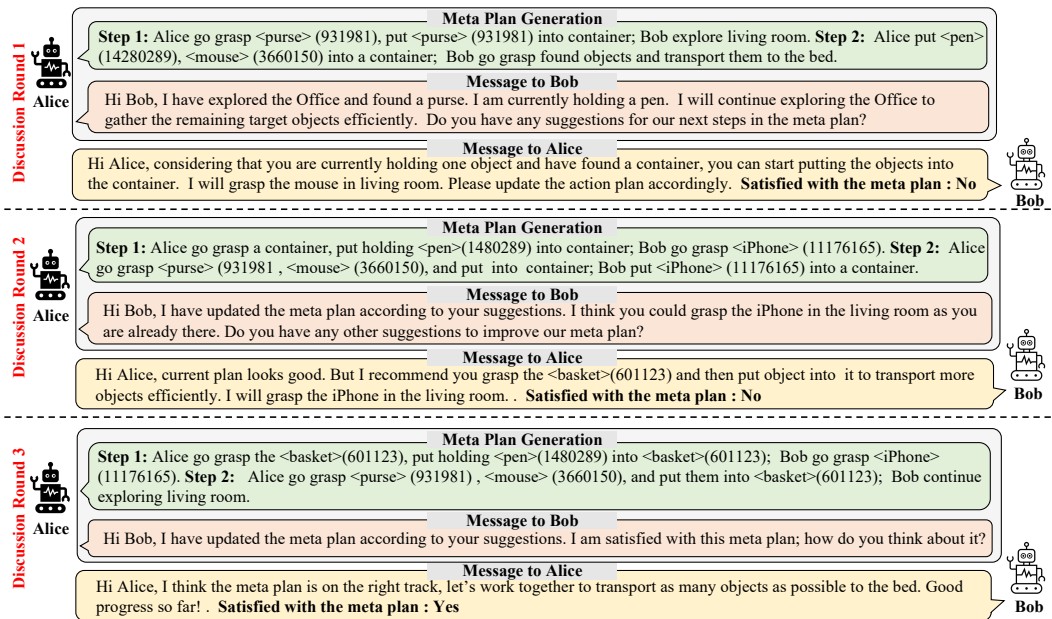

Figure 3: **Examples of the evaluation and optimization process of meta-plan via multi-turn discussion between agents**. The discussion is triggered by new progress, i.e., Alice founds new object 'purse'. Here, Alice acts as the meta-plan designer, while Bob serves as the meta-plan evaluator. The example is derived from the transporting task of TDW-MAT.

As shown in Fig. 2, each agent is equipped with a communication module powered by a pretrained LLM to facilitate multi-turn discussions. Specifically, the communication module first retrieves relevant information from the memory, e.g., meta-plan, agent state, and previous dialogue history among agents, then prompts the LLM to generate the message to send via the following prompt:

> Prompt: `<Task Desc>` + `<Instruct Head>` + `<Meta-plan>` + `<Agent State>` + `<Dialog History>` \n.        LLM: `<Messages>`.

The tags `<Meta-plan>`, `<Agent State>`, and `<Dialog History>` act as placeholders for inserting the meta-plan, the agent's state, and the dialogue history between agents. The tag `<Instruct Head>` differs for the meta-plan designer and evaluator: the former instructs the LLM to generate messages asking teammates for their opinions, while the latter focuses on providing feedback on the meta-plan. After receiving feedback from the teammate agents, the meta-plan creator reinitiates the process to generate a new meta-plan. Fig. 3 illustrates the evaluation and optimization process of a meta-plan through multi-turn discussions among agents. It is evident that the optimized meta-plan effectively integrates partially observed information from all agents, resulting in improved coordination and efficiency. Detailed prompt designs for the communication module can be found in Fig. 10 and 11 in the Appendix.

## 4.3 Progress-Adaptive meta-plan & Execution

The optimized meta-plan acts as a high-level guide, assigning subtasks to each agent and coordinating them for efficient task completion. However, due to dynamic environmental changes and task progress updates, the meta-plan can become outdated during execution. As illustrated in Fig. 4, agents may encounter significant progress, such as discovering target objects or completing subtasks, necessitating adjustments to the meta-plan. In such cases, the previous plan becomes less effective or invalid for coordinating the agents.

To address this, we design a *progress-adaptive planning module* for CaPo for adapting the meta-plan to the agents' latest progress. This module follows a similar process as described in Sec. 4.2—meta-plan initialization, evaluation, and optimization—but with modified prompting strategies for the

LLMs. Whenever an agent makes new progress, the meta-plan designer promptly generates an updated meta-plan, followed by a multi-turn discussion among all agents to further optimize it. The LLM prompting strategies for the progress-adaptive planning module are structured as follows:

> Prompt: `<Task Desc>` + `<Instruct Head>` + `<Meta-plan>` + `<Agent Progress>` + `<Teammate Progress>` + `<Dialog History>` `\n`.
> LLM: `<meta-plan>` or `<Messages>`.

Here we introduce two placeholders, `<Agent Progress>` and `<Teammate Progress>`, to capture the task progress of agents and enable the LLM to generate progress-aware responses, such as meta-plans or communication messages. Agents engage in discussions to optimize the meta-plan until a consensus is reached or communication resources are exhausted (e.g., after three discussion rounds). Detailed prompt designs for the LLMs—responsible for generating the meta-plan and facilitating messages for both the meta-plan designer and evaluator—are provided in Fig. 11~12.

Once the meta-plan or progress-adaptive meta-plan is established, each agent autonomously transforms the plan into executable actions via a plan parsing module and an execution module. The plan parsing module generates the latest sub-plan by retrieving relevant information from the memory module and converting it into text descriptions, and then compiles an Action List of all available high-level sub-plans. We implement the plan parsing module as a pretrained LLM, and prompt it with a concatenation of `Instruct Head`, `Task Description`, `meta-plan`, `Action History`, `Agent Progress`, and `Action List` to choose the most suitable sub-plan. See

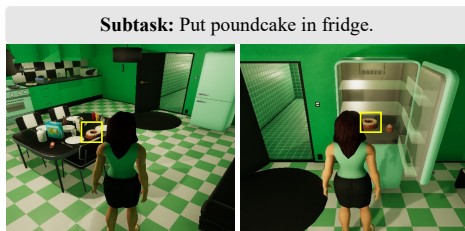

**Subtask:** Put poundcake in fridge.

(a) Discover a target object  (b) Complete a subtask

Figure 4: **Two types of new progress during task execution.** (a) Discover a new object poundcake. (b) complete a subtask.

Fig. 13 in Appendix for more prompt details. Given the sub-plan, we adopt a similar execution module as in (Zhang et al., 2023b) to generate primitive actions for executing the sub-plan.

## 5 EXPERIMENTS

**Benchmarks.** We follow CoELA, and adopt the ThreeDworld Multi-Agent Transport (TDW-MAT) task (Zhang et al., 2023b), and the Communicative Watch-And-Help (C-WAH) task (Zhang et al., 2023b) to test our CaPo. TDW-MAT is built on the general purpose virtual world simulation platform TDW platform (Gan et al., 2020), and requires agents to move objects by their hands or containers which can contain several objects for efficient moving to the destination. Moreover, agents can receive ego-centric $512 \times 512$ RGB-D images as observation and can communicate with others. The test set of TDW-MAT consists 24 episodes, which evenly divided into food and stuff tasks. In C-WAH, agents are requested to complete five types of household activities, represented as various predicates with specific counts that must be satisfied. The test set contains 10 episodes, including both *symbolic and visual observation settings*. More details about TDW-MAT and C-WAH environments are provided in Appendix B.1 and B.2, respectively.

**Metrics.** On TDW-MAT, we adopt *Transport Rate*, i.e., the fraction of subtasks completed within 3000 time steps (a.k.a. frames), as performance metric. Note, one action step may last multiple time steps, e.g., resetting arms. On C-WAH, *Average Steps* to complete all tasks is used as the metric to evaluate cooperation efficiency. Following CoELA (Zhang et al., 2023b), we also report *Efficiency Improvement (EI)* of cooperating with other agents as $\Delta M/M_0$, where $\Delta M$ is the main efficiency metric difference, and $M_0$ denotes the larger on of the main efficiency metric for numerical stability.

**Implementation.** We test two settings on TDW-MAT task: 1) a real-world setting where the perception module is instantiated as Mask-RCNN (He et al., 2017) that is trained using collected scene images (Zhang et al., 2023b), and 2) an oracle setting with segmentation ground-truth. We use GPT-3.5-turbo and GPT-4 from the OpenAI API (OpenAI, 2024), and LLAMA-2-13B-CHAT (Touvron et al., 2023), as LLMs in embodied agents. We set default parameters for LLMs: temperature of 0.7, a maximum of 256 output tokens, and top-1 sampling. Our code will be made publicly available.

| | Classic Agents | | GPT-3.5 Agents | | LLAMA-2 Agents | | GPT-4 Agents | | | |
|---|---|---|---|---|---|---|---|---|---|---|
| | RHP*† | RHP† | CoELA | CaPo(ours) | CoELA† | CaPo(ours) | CoELA† | ProAgent | RoCo | CaPo(ours) |
| **w/o Oracle Perception** | | | | | | | | | | |
| Food (↑) | 49 | 67 +25% | 67 +23% | 70 +31% | 57 +9% | 66 +17% | 82 +38% | 82 +27% | 83 34% | 85 +43% |
| Stuff (↑) | 36 | 54 +34% | 39 +18% | 45 +27% | 48 +11% | 56 +22% | 61 +41% | 57 +35% | 60 +39% | 64 +40% |
| Avg. (↑) | 43 | 61 +29% | 52 +20% | 57 +29% | 53 +10% | 61 +19% | 71 +39% | 69 +31% | 71 +36% | 74 +41% |
| **w/ Oracle Perception** | | | | | | | | | | |
| Food (↑) | 52 | 76 +33% | 72 +29% | 85 +38% | 60 +3% | 66 +14% | 87 +41% | 84 +37% | 88 +42% | 90 +40% |
| Stuff (↑) | 49 | 74 +34% | 73 +32% | 84 +39% | 63 +19% | 76 +23% | 83 +41% | 85 +34% | 82 +35% | 87 +38% |
| Avg. (↑) | 50 | 75 +34% | 72 +30% | 84 +38% | 62 +8% | 71 +18% | 85 +41% | 84 +35% | 85 +38% | 89 +39% |

Table 1: **Comparison of average Transport Rate(%) of all baselines on the TDW-MAT w/o and w/ Oracle Perception task.** Each task requires agents to move two kinds of items, including Food and Stuff. RHP* uses a single agent while all others adopt two agents. † denotes results quoted from CoELA. The subscript value like +25% in 67 +25% denotes the *Efficiency Improvement*.

| | Classic Agents | | Heterogeneous Agents | | GPT-4 Agents | | | |
|---|---|---|---|---|---|---|---|---|
| | MHP*† | MHP† | MHP+CoELA† | MHP+CaPo | CoELA† | ProAgent | RoCo | CaPo(ours) |
| Symbolic Obs (↓) | 111 | 75 +33% | 59 +45% | 57 +47% | 57 +49% | 62 +37% | 57 +43% | 51 +46% |
| Visual Obs (↓) | 141 | 103 +26% | 94 +34% | 90 +38% | 92 +34% | 90 +37% | 89 +32% | 83 +37% |

Table 2: **Comparison of Average Steps of all methods on the C-WAH task.** "Symbolic Obs" and "Visual Obs" denote symbolic and visual observation settings, respectively. MHP* uses a single agent while all others adopt two agents. † indicates results quoted from CoELA. The subscript value like +33% in 75 +33% denotes the *Efficiency Improvement*.

**Baselines.** We adopt three types of methods as our baseline: 1) classical agents, including MCTS-based Hierarchical Planner (**MHP**) (Puig et al., 2020) and Rule-based Hierarchical Planner (**RHP**) (Gan et al., 2022). 2) LLM-driven agents, including CoELA Zhang et al. (2023b), ProAgent (Zhang et al., 2023a), and RoCo (Mandi et al., 2023). CoELA Zhang et al. (2023b) features a modular framework for multi-agent planning, communication, and complete long-horizon tasks, but generate independent short-term plan for each agent. ProAgent (Zhang et al., 2023a), and RoCo (Mandi et al., 2023) generate joint plans for cooperative agents, and introduce a reflection loop or environment feedback for plan validation. 3) Heterogeneous agents, where agents with different planning and communication mechanisms collaborate within the same environment. See more details in Appendix A.1 and A.2.

## 5.1 MAIN RESULTS

**Performance comparison.** We follow CoELA to test two-agent cooperation setting, and compare with classical methods like MHP and RHP, and LLM-driven methods CoELA, ProAgent, and RoCo. Table 1 summarizes the performance of all compared methods under the two settings of the TDW-MAT task, and shows several observations. **1)** Compared with the single-agent baseline RHP, CaPo and all two-agent baselines consistently make significant improvements, showing the effectiveness of multi-agent cooperation in embodied tasks. **2)** In multi-agent comparisons, our CaPo outperforms LLM-driven methods by a clear margin, e.g., respectively making 4.7% and 5.9% improvement over CoELA and RoCo under the oracle perception setting. **3)** CaPo demonstrates consistently superior performance across all settings when paired with different LLMs as the agent brain. Notably, even with a less advanced LLM like GPT-3.5-turbo, CaPo achieves a significantly higher performance compared to the baseline CoELA. For example, under the oracle perception setting, CaPo achieves a transport rate of 84, outperforming CoELA's 72. The improvement of CaPo is derived from its meta-plan and progress-adaptive meta-plan, which both provide strategial and coherent guidance for agent cooperation, thereby improving cooperation performance.

Table 2 reports the performance of all methods on the C-WAH task. Specifically, with GPT-4 agents, our CaPo respectively makes 9.8%, 7.6% and 6.6% relative improvement on CoELA, ProAgent and RoCo. Interestingly, CaPo, when paired with MHP, where the CaPo agent independently creates and updates the meta-plan, outperforms both a team of two MHP agents and a combination of MHP

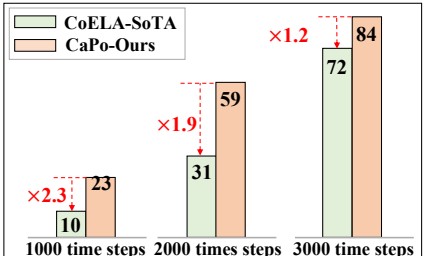

Figure 5: **Comparison of Transport Rate (%) of CoELA and CaPo using GPT-3.5 under different time steps**.

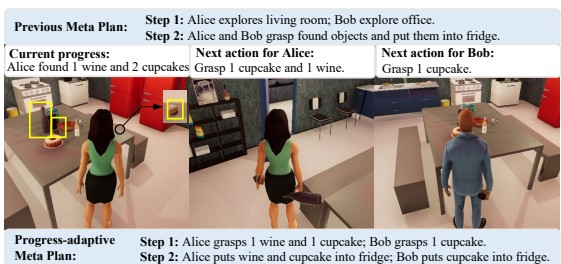

Figure 6: **Example of progress-adaptive meta-plan adaptation.** New progress: Alice found target objects, 1 wine and 2 cupcakes.

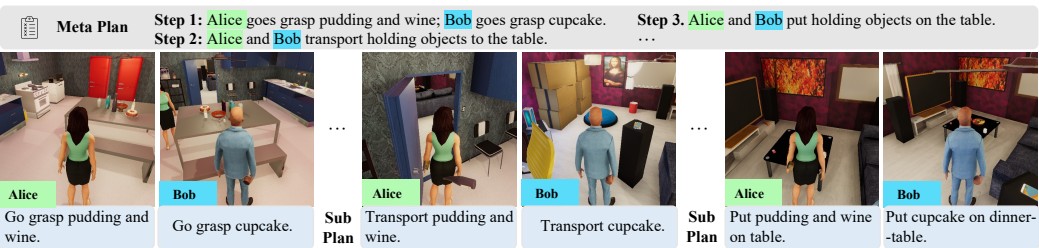

Figure 7: **Examples of cooperative behaviors introduced by meta-plan.** Guided by meta-plan, agents show clear work and task allocation, thereby improving cooperative efficiency.

and CoELA agents. These results consistently highlight the superiority of CaPo in enhancing multi-agent cooperation. We provide more analysis on heterogeneous agents in Appendix A.2.

**Efficiency comparison.** To demonstrate the cooperation efficiency, we compare the transport rates of CaPo and CoELA with different time steps on TDW-MAT. As shown in Fig. 5, with the same time step budget and GPT-3.5 agents, CaPo consistently outperforms CoELA across various time steps, indicating its superiority to coordinate agents effectively. The improvement is particularly clear in scenarios of small time steps. For example, given 1,000 time steps, CaPo doubles the transport rate of CoELA by improving 10% to 23%. This shows that with limited time or resources, a cooperative meta-plan can significantly improve cooperation efficiency.

**Qualitative Analysis.** Here we investigate the agents' behavior in CaPo with GPT-4 on the C-WAH task. In the meta-plan generation phase, as shown in Fig. 3, agents ask questions, provide feedback, and collaboratively refine the initial meta-plan. Moreover, in this phase, Fig. 7 shows that with meta-plan as guidance, two agents, Alice and Bob, have clear work/labor allocation to complete tasks, thereby avoiding redundant steps and improving cooperation efficiency. For the progress-adaptive meta-plan and execution phase, Fig. 6 also shows that when agent Alice achieves progress, e.g., discovering three target objects, both agents will accordingly discuss to adapt the meta-plan, e.g., grasping 1 wine and 1 cupcake by Alice. This ongoing adaptation of the meta-plan provides strategic, coherent, and timely guidance, facilitating efficient coordination among agents and ultimately enhancing multi-agent cooperation.

## 5.2 ABLATION STUDY

**Effects of each component in CaPo.** Here we examine the effects of two key components: 1) meta-plan generation, which includes meta-plan initialization, evaluation, and optimization, and 2) the progress-adaptive meta-plan. To evaluate their impact, we first remove both components from CaPo, resulting in $CaPo_1$. As shown in Table 3, $CaPo_2$, which includes meta-plan initialization but freezes the meta-plan during subsequent procedures, improves upon $CaPo_1$ by approximately 1% across three metrics, demonstrating the value of meta-plan initialization. Similarly, $CaPo_3$, which incorporates the full meta-plan generation process, outperforms $CaPo_2$ by a significant margin, highlighting the benefits of meta-plan evaluation and optimization. Finally, CaPo achieves a 7% improvement over $CaPo_3$, showcasing the effectiveness of the progress-adaptive meta-plan. These results underscore the importance of each component in the CaPo framework.

| Method | Food (↑) | Stuff (↑) | Avg. (↑) |
|---|---|---|---|
| **CaPo$_1$** (No MP + No Pro. MP) | 72 | 75 | 73 |
| **CaPo$_2$** (MP Initialization + No Pro. MP) | 73 | 76 | 74 |
| **CaPo$_3$** (MP Generation + No Pro. MP) | 74 | 80 | 77 |
| **CaPo** (MP Generation + Pro. MP) | **85** | **84** | **84** |

Table 3: **Effects of the components in CaPo using GPT-3.5 on TDW-MAT task**. We report the transport Rate (TR, %). MP" denote 'Meta Plan" and Progress-Adaptive Meta Plan", respectively.

| Method | Symbolic Obs (↓) | Visual Obs (↓) |
|---|---|---|
| **CaPo×1** | 93 | 106 |
| **CaPo×2** | 51 | 83 |
| **CaPo×3** | 46 | **72** |
| **CaPo×4** | **45** | 74 |

Table 4: **Benefits of increasing agent numbers in our CaPo using GPT-4 on the C-WAH task**. Average steps required to complete task are reported.

**Effects of agent number.** Table 4 investigates the effects of agent number in CaPo using GPT-4 on the C-WAH task, where "CaPo × C" denotes using C GPT-4 agents. We can observe that increasing agent number to 3 significantly reduces the overall time step number required to complete tasks. This improvement also shows the effectiveness of our CaPo on multiple agent cooperation. However, increasing the number of agents to four results in only minor or degraded improvements. This is because for simple tasks, agents are too much and suffer from highly-frequent agent dispatch, leading to inferior collaboration efficiency. For instance, setting up a dining table does not require four waiters, as a maximum of two agents is sufficient.

**Progress in meta-plan adaptation.** Fig. 4 in Sec. 4.3 shows two clear task progress examples: 1) discovering a target object and 2) completing a subtask, both of which can trigger agents to adapt the meta-plan to their latest task progress. Such progress is crucial, as agents need to continually refine the meta-plan to complete tasks efficiently and maximize cooperation. Conversely, actions without significant progress, such as entering a new room, do not prompt agents to adjust the current (progress-adaptive) meta-plan. This is because it would be unnecessary, and updating the meta-plan involves communication, which incurs additional time overhead.

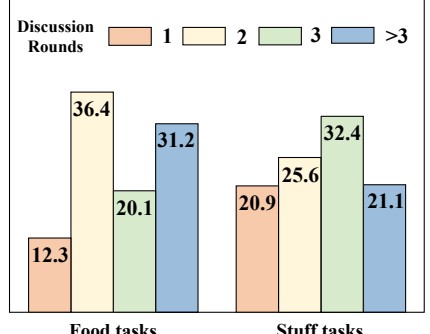

Figure 8: **Percentage (%) of discussion rounds needed for agents to reach consensus on a meta-plan**, based on results from TDW-MAT.

**Convergence analysis of agent discussion.** Here we investigate the convergence of agent discussions, focusing on how many rounds are required for agents to reach consensus on the meta-plan. In the TDW-MAT environment, we limit the maximum number of discussion rounds—referred to as the discussion budget—to three (details in Appendix A.6). As shown in Fig. 8, agents reach consensus within three rounds in most cases, with 78.9% achieving consensus in the "Stuff" tasks. By capping the discussion rounds, CaPo achieves a balance between discussion effectiveness and the budget, avoiding unnecessary or prolonged deliberations. Additionally, we provided an communication analysis in Appendix A.7, which demonstrate CaPo achieves a good balance between cooperation efficiency and communication cost.

## 6 CONCLUSION AND LIMITATIONS

In this work, we introduce Cooperative Plan Optimization (CaPo) to improve the cooperation efficiency of LLM-driven embodied agents. CaPo creates a strategic and coherent meta-plan through multi-turn agent discussions before any actions are executed. This meta-plan acts as an action guide to effectively coordinate multiple agents in task completion. During execution, agents dynamically adapt the meta-plan based on task progress, ensuring its continued effectiveness in coordination. Experimental results on TDW-MAT and C-WAH demonstrate that CaPo achieves higher task completion rates and efficiency compared to state-of-the-art methods.

**Potential Limitation:** While CaPo significantly improves multi-agent cooperation efficiency, it has certain limitations. Specifically, it relies heavily on LLMs for reasoning and planning during meta-plan generation and adaptation. Additionally, CaPo employs a pre-defined communication protocol for creating and updating the meta-plan, which may restrict its flexibility in dynamic or heterogeneous environments. Addressing these limitations will be a focus of our future work.

## ACKNOWLEDGMENTS

This work was partially funded by Elekta Oncology Systems AB and a RVO public-private Partnership grant (PPS2102).

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

# Appendix

## A ADDITIONAL EXPERIMENTS AND DISCUSSION

### A.1 BASELINE MODELS AND OTHER ENVIRONMENTS

We adopt two types of methods as our baseline, including classical agents and LLM-driven multi-agents. (1) The classical agents include MCTS-based Hierarchical Planner (**MHP**) (Puig et al., 2020) which is a hierarchical planner originating from the original Watch-And-Help Challeng, and Rule-based Hierarchical Planner (**RHP**) (Gan et al., 2022) derived from a strong baseline in the ThreeDWorld Transport Challenge. (2) LLM-driven agents consist of CoELA Zhang et al. (2023b), ProAgent (Zhang et al., 2023a), and RoCo (Mandi et al., 2023). Cooperative Embodied Language Agent (**CoELA**) (Zhang et al., 2023b) can plan, communicate, and collaborate with other agents to complete long-horizon tasks, but generate independent short-term plan for each agent. In addition, we also introduce two more baselines – ProAgent (Zhang et al., 2023a) and RoCo (Mandi et al., 2023), and implement them on TDW-MAT and C-WAH using source codes. These two baselines generate joint plans for cooperative agents, and introduce a reflection loop or environment feedback for plan validation.

We conducted experiments on TDW-MAT and C-WAH, which are widely recognized benchmarks for embodied multi-agent cooperation. While other environments, such as ALFRED (Shridhar et al., 2020) and BEHAVIOR-1K (Li et al., 2023a), are notable in this domain, they do not support multi-agent cooperation simulation, and therefore, we did not include them in our experiments. Similarly, works based on Minecraft (Wang et al., 2023b; Cai et al., 2023), such as (Wang et al., 2023c;d), focus on settings that differ from ours, as our framework emphasizes decentralized cooperation with partial observation. Consequently, Minecraft-based environments were also excluded from our experiments.

### A.2 EXPERIMENTS ON HETEROGENEOUS AGENTS' COOPERATION

CaPo introduces a pre-defined communication protocol to create and update the meta-plan, which facilitate the multi-cooperation efficiency. However, it is also very interesting to explore whether CaPo could improve cooperation efficiency when cooperating with other decentralized agents, e.g, RHP and CoELA. Specifically, when cooperating with other agents, the CaPo agent independently creates and update the meta-plan due to the lack of multi-round discussion. We conduct experiments on the TDW-MAT tasks using oracle perception, and the results are shown in Table 5.

| Runs | Food (↑) | Stuff (↑) | Avg. (↑) |
|------|----------|-----------|----------|
| RHP[†] | 0.52 | 0.49 | 0.50 |
| RHP+RHP[†] | 0.76 (+33%) | 0.74(+34%) | 0.75(+33%) |
| RHP+CoELA[†] | 0.85 (+40%) | 0.77 (+35%) | 0.81 (+37%) |
| RHP+CaPo | 0.85 (+38%) | 0.80 (+37%) | 0.82 (+37%) |
| CaPo+CoELA | 0.86 (+36%) | 0.83 (+35%) | 0.84 (+35%) |

Table 5: **Results on heterogeneous agents' cooperation.** Here we report transport rate (efficiency improvement) on TDW-MAT tasks using oracle perception and GPT-4. [†] denotes results from the CoELA paper. The reported EI may fluctuate even for same transport rate, due to slight variations of the performance of baseline agent.

As shown in the above table, when CaPo cooperates with RHP agents, it achieves a higher transport rate compared to both a team of two MHP agents and a combination of MHP and CoELA agents. We attribute this improvement to the independently developed meta-plan, which guides the CaPo agent to complete tasks more efficiently and implicitly enhances multi-agent cooperation efficiency.

Interestingly, when our CaPo cooperates with the more advanced agent CoELA, it further improves the transport rate. This highlights the potential of deploying CaPo to collaborate with other decentralized agents.

## A.3 REPRODUCIBILITY OF RESULTS

LLM-driven reasoning and planning tend to be stochastic, requiring multiple runs to assess stability. To verify the stability and reproducibility of our method, we conducted three runs on TDW-MAT with oracle perception and GPT-3.5 agents. As shown in Table 6, the results exhibit minor variance across runs, demonstrating the stability and reproducibility of our method.

| Runs | Food ($\uparrow$) | Stuff ($\uparrow$) | Avg. ($\uparrow$) |
|---|---|---|---|
| 1 | 0.85 | 0.84 | 0.84 |
| 2 | 0.84 | 0.81 | 0.82 |
| 3 | 0.85 | 0.83 | 0.84 |
| Average | 0.84 (0.006) | 0.82 (0.015) | 0.83 (0.012) |

Table 6: **Transport Rate (TR) comparison on the TDW-MAT task using GPT-3.5 and oracle perception.** We perform 3 runs with random seeds and report mean and variance.

## A.4 RESULTS ON ADDITIONAL ENVIRONMENTS

To test the generalization and effectiveness of our model on new environments, we add one more experiments on RoCoBench (Mandi et al., 2023), which is a multi-robot collaboration environment. We choose three tasks exhibiting asymmetric observation, which are more relevant to decentralized, partia-observation settings, and the results are shown in the Table 7. Our CaPo framework achieves a higher success rate with fixed steps and requires few steps to complete the task. This further highlights the effectiveness and generalization of our model on embodied multi-agent cooperation.

|  | Sweep Floor | | Pack Grocery | | Move Rope | |
|---|---|---|---|---|---|---|
|  | Success rate | Step | Success rate | Step | Success rate | Step |
| **RoCo** | $0.95 \pm 0.05$ | 7.1 | $0.44 \pm 0.06$ | 9.9 | $0.65 \pm 0.11$ | 2.5 |
| **CaPo (ours)** | $0.98 \pm 0.13$ | 6.5 | $0.63 \pm 0.11$ | 7.6 | $0.74 \pm 0.17$ | 2.2 |

Table 7: Efficiency comparison on RocoBench (Mandi et al., 2023). We report two metrics, success rate of the task, and average steps required to complete the task.

## A.5 NUMBER OF DISCUSSION ROUND

To generate or update the meta plan, all agents are involved in multi-round discussion. We set the discussion budget to 3 rounds to balance the consensus rate and overall efficiency. As shown in the Table A.5, with a 3-round limit, the failure rate decreases to 21.1%, and overall efficiency improves to 84%. Extending the discussion budget beyond 3 rounds offers only marginal benefits in reducing failed cases (e.g., a reduction to 17.8% with 4 rounds) while providing no additional efficiency gains, as efficiency stabilizes at 84%. These results demonstrate that a 3-round discussion budget is a reasonable choice, effectively balancing communication costs and task performance.

| Maximal Round | 1 | 2 | 3 | 4 | 5 |
|---|---|---|---|---|---|
| Rate of failed cases (%) | 79.1 | 53.5 | 21.1 | 17.8 | 15.5 |
| Transport rate (%) | 79 | 82 | 84 | 84 | 83 |

Table 8: Failed case rate and transport rate under different maximal rounds of discussion.

## A.6 COMMUNICATION COST

For communication cost and efficiency, we want to discuss them from several aspects. Firstly, the task cost mainly includes agent moving cost and communication cost. By comparison, moving cost

is often much higher than communication cost, since current robots lack the ability of humans, and their movement is costly especially in complex and large-scale environments like urban areas; while the latter only needs LLM's inference, and massage sending and receiving, and is indeed much easier and efficient. For example, moving from one room to another can often require much more time than LLM's inference.

Moreover, to reduce communication cost, our CaPo adopts two strategies. a) agent's progress-adaptive planning module will evaluate whether meta plan is suitable for current task progress: if yes, then all agents will continue their subtasks, and will not communicate, reducing uncommunica-tion costs; b) with the meta plan as a reference, a few round communication is sufficient to arrive at a new plan, since task progress is often smoothly achieved and thus adaptation the meta plan need slightly small changes. Indeed, we set the maximum round of communication as 3 for each discussion. So these strategies differ from CoELA which needs communication at each iteration, while ours does not.

| Method | # Rounds of Meta-plan Updates | Character Cost per Meta-plan | Frames Cost from Communication | # Rounds of Discussion per Meta-plan | Time Cost per Meta-plan Update | Time Cost per Task |
|---|---|---|---|---|---|---|
| **Food tasks** | 14 | 1,757 chars | 56 frames | 2.4 rounds | 20.4 sec | 26.2 min |
| **Stuff tasks** | 17 | 1,462 chars | 51 frames | 2.1 rounds | 22.7 sec | 25.6 min |

Table 9: Communication cost analysis on the TDW-MAT dataset with oracle perception.

For average rounds of meta-plan updates and average steps for each meta plan discussion, we report the statistics Table 9. Regarding Food tasks on TDW, agents update the meta-plan for an average of 14 times per task, with each discussion requiring 2.4 steps and a total 1,757 characters for sending (i.e., 4 frames since 1 frame can send 500 characters as in CoELA). While this incurs some time cost, the long-term coherent meta-plan results in a substantial transport rate gain. This demonstrates that creating a coherent meta-plan through communication is significantly more time-efficient than relying solely on embodied actions.

## A.7 COLLABORATING WITH HUMAN

To evaluate human-agent cooperation, we follow the methodology of CoELA and conduct experiments on the C-WAH tasks, where the agent Bob (meta-plan evaluator) is controlled by real human participants. We adopt average steps as an efficiency metric and also ask three participants to rate their teammate agent (i.e., CaPo or CoELA) on a 7-point Likert Scale based on three criteria: communication effectiveness, helpfulness, and trust. The results are detailed in Table A.7.

| Method | Average Steps | Comm Effectiveness | Helpfulness | Trust |
|---|---|---|---|---|
| **CoELA** | $47 \pm 2$ | $5.8 \pm 0.3 / 7$ | $5.5 \pm 0.4 / 7$ | $6.0 \pm 0.1 / 7$ |
| **CaPo (ours)** | $44 \pm 1$ | $6.3 \pm 0.1 / 7$ | $6.1 \pm 0.3 / 7$ | $6.3 \pm 0.2 / 7$ |

Table 10: Comparison of CoELA and CaPo on Human Collaboration Metrics.

**Efficiency (Average steps).** Humans collaborating with CaPo generally achieve higher efficiency, completing tasks with fewer steps, i.e., 44 steps. This is because both agents and humans can avoid redundant actions through the guidance of the proposed meta-plan.

**Human Preferences.** (1) **Communication effectiveness** represents the effectiveness of communication between agent and human participants, e.g., whether the agent understands the message and provides useful information. In general, CaPo demonstrates clearer and more reliable communication than CoELA (6.3 vs. 5.8). This improvement is attributed to CaPo's multi-round discussion mechanism, which creates a long-term, coherent meta-plan, reducing the need for frequent communication. In contrast, in CoELA, agents and humans have to exchange information more frequently, due to the lack of a coherent plan. Finally, agents didn't show verbose behavior when making plans (as shown in Figure 3), as we constrained the LLM to generate concise plans and messages in prompting. (2) **Helpfulness** indicates whether the agent helps human participants complete the task faster. By multi-round discussion and creating coherent meta-plans, CaPo could help both agent

and human participants to reduce redundant actions in the embodied environments, e.g., aimless exploration. (3) **Trust** measures how human participants trust their agent teammate. Humans show a stronger preference for trusting CaPo (6.3 vs. 6.0) during the cooperation. This is because CaPo involves humans in the planning process, seeking their input during multi-round discussions to collaboratively create the meta-plan. In contrast, CoELA agents act more independently and sometimes fail to respond to human messages, which weakens trust.

Through human-agent cooperation experiments, we demonstrate that CaPo enhances communication clarity, actively supports human decision-making, and fosters trust by engaging humans in the planning process. These advantages position CaPo as a more favorable teammate for human collaborators, underscoring its potential for real-world cooperative applications.

# B  ADDITIONAL ENVIRONMENT DETAILS

We evaluate our method and all baseline methods in two simulated environments: ThreeDWorld Multi-Agent Transport (TDW-MAT) and Communicative Watch-And-Help (C-WAH). We follow CoELA Zhang et al. (2023b) and list detailed introductions to these environments below.

## B.1  THREEDWORLD MULTI-AGENT TRANSPORT

**Tasks.**     TDW-MAT consists of two types of tasks, food-transporting task and stuff-transporting task. The food-transporting task has 6 types of targets (apple, banana, orange, bread, loaf bread, and burger) and 3 containers (bowl, plate, and tea tray). In contrast, the stuff-transporting task has 6 different types of targets(calculator, mouse, pen, lighter, purse, and iPhone) and 3 containers (plastic basket, wood basket, and wicker basket). In each task, there are 10 target objects and 2 to 5 containers in total. Additionally, there are 4 types of rooms: living room, office, kitchen, and bedroom, and objects are placed in these rooms consistent with common sense. The agents are tasked with transporting as many target objects as possible to the goal position using containers as tools. Each container can carry up to three objects, while without a container, an agent can transport only two objects at a time. The agents must transport as many target objects as possible within 3000 frames.

**Observation Space**     The embodied agent receives a variety of observations, with the primary ones being an egocentric RGB image and a depth image. Additionally, there are several auxiliary observations. The observation space includes:

- **RGB image:** This is an egocentric image captured by a forward-facing camera, with a resolution of $512 \times 512$ and a field of view of 90 degrees.
- **Depth image:** This image shares the same camera intrinsic parameters as the RGB image.
- **Oracle Perception (optional):** An image where each object ID is represented by a distinct color, using the same camera intrinsic parameters as the RGB image.
- **Agent position and rotation**: The position and rotation of the agent within the simulation environment.
- **Messages**: Communications sent by all agents.
- **Held objects**: Information about the objects currently held by the agent.
- **Opponent held objects**: Information about objects held by another agent, if the agent is within view.

**Action Space**     In TDW-MAT, agents can perform 7 distinct types of actions to interact with the environment or communicate with each other. Each action spans multiple frames, and the detailed action space is outlined below:

- **Move forward**: The agent advances by 0.5m.
- **Turn left**: The agent rotates left by 15 degrees.
- **Turn right**: The agent rotates right by 15 degrees.

- **Grasp**: The agent grasps an object, successfully performing this action only when in close proximity to the object. The object can be either a target or a container.

- **Put In**: The agent places a target into a container, an action that is possible only when the agent is holding a target in one hand and a container in the other.

- **Drop**: The agent releases the objects held in hand.

- **Send message**: The agent sends a message to other agents, with a limit of 500 characters per frame.

## B.2 COMMUNICATIVE WATCH-AND-HELP

Communicative Watch-And-Help (C-WAH) builds upon the Watch-And-Help challenge Puig et al. (2021) by incorporating the ability for agents to send messages to one another. Sending messages, like other actions, consumes one timestep and is subject to a maximum length constraint.

| Task Name | Predicate Set |
|---|---|
| Prepare afternoon tea | ON(cupcake,coffeetable), ON(pudding,coffeetable), ON(apple,coffeetable), ON(juice,coffeetable), ON(wine,coffeetable) |
| Wash dishes | IN(plate,dishwasher), IN(fork,dishwasher) |
| Prepare a meal | ON(coffeepot,dinnertable),ON(cupcake,dinnertable), ON(pancake,dinnertable), ON(poundcake,dinnertable), ON(pudding,dinnertable), ON(apple,dinnertable), ON(juice,dinnertable), ON(wine,dinnertable) |
| Put groceries | IN(cupcake,fridge), IN(pancake,fridge), IN(poundcake,fridge), IN(pudding,fridge), IN(apple,fridge), IN(juice,fridge), IN(wine,fridge) |
| Set up a dinner table | ON(plate,dinnertable), ON(fork,dinnertable) |

Table 11: **Task description in C-WAH**. The tasks are divided into five types, each containing several predicates.

**Tasks** The Communicative Watch-And-Help (C-WAH) framework includes five types of tasks: *Prepare afternoon tea*, *Wash dishes*, *Prepare a meal*, *Put groceries*, and *Set up a dinner table*. These tasks encompass various household activities, each consisting of several subtasks described by predicates in the "*ON/IN(x, y)*" format, such as "*Put x ON/IN y*". Detailed descriptions of the tasks are provided in Table 11.

The primary objective is to complete all given subtasks within 250 timesteps, with each task containing between 3 to 5 subtasks.

**Observation Space** C-WAH offers two modes of observation: *Symbolic Observation* and *Visual Observation*.

In *Symbolic Observation*, following the original Watch-And-Help challenge, an agent can access comprehensive object information within the same room, including location, status, name, and relationships.

In *Visual Observation*, agents receive egocentric RGB and depth images along with auxiliary observations. The detailed observation space includes:

- **RGB image:** An egocentric image from a forward-facing camera, with a resolution of $256 \times 512$ and a field of view of 60 degrees.

- **Depth image:** An image with the same camera intrinsic parameters as the RGB image.

- **Oracle Perception:** An image where each object ID is mapped to a color, sharing the same camera intrinsic parameters as the RGB image.
- **Agent position:** The agent's position within the simulation world.
- **Messages**: Communications sent by all agents.
- **Held objects**: Information about the objects currently held by the agent.
- **Opponent held objects**: Information about objects held by another agent, if visible.

**Action Space**    The action space in C-WAH closely resembles that of the original Watch-And-Help Challenge, with the addition of the *send message* action. The detailed action space includes:

- **Walk towards**: Move towards an object in the same room or towards a specific room.
- **Turn left**: Rotate left by 30 degrees.
- **Turn right**: Rotate right by 30 degrees.
- **Grasp**: Grasp an object, which can be successfully performed only when the agent is close to the object.
- **Open**: Open a closed container, performable only when the agent is near the container.
- **Close**: Close an open container, performable only when the agent is near the container.
- **Put**: Place held objects into an open container or onto a surface, performable only when the agent is near the target position.
- **Send message**: Communicate with other agents, with a limit of 500 characters per message.

## C  PROMPT TEMPLATE

We list the prompts template for meta plan initialization, communication module of Alice, communication module of Bob, cooperative planning module, and the plan parsing module as follows.

---

I am Alice. My teammate Bob and I want to transport as many target objects as possible to the bed with the help of containers within 3000 steps. I can hold two things at a time, and they can be objects or containers. I can grasp containers and put objects into them to hold more objects at a time.

Assume that you are an expert plan outline designer. Given our shared goal, please help me generate a global meta plan for me and Bob during task execution, guiding me and Bob to achieve the goal collaboratively as soon as possible. Note that a container can contain three objects, and will be lost once transported to the bed. I can only put objects into the container I hold after grasping it. All objects are denoted as <name> (id), such as <table> (712). Actions take several steps to finish. It may be costly to go to another room or transport to the bed, use these actions sparingly.

The generated meta plan must meet following requirements:

1.There are 5 allowed actions you can use to construct the meta plan. 1) 'go to': move to a specified room. 2) 'explore': explore a room for underlying target objects. 3) 'go grasp': go to grasp a specified target object. 4) 'put': Place an object into a specified container. 5) 'transport': Transport holding objects or containers to the bed and drop them on the bed.

2.The meta plan should be concise, brief, and reliable.

3.The meta plan must be structured strictly in the three-step format: {Action Plan: Step 1: Alice xxx, Bob xxx; Step 2: Alice xxx, Bob xxx; Step 3: Alice xxx, Bob xxx}. Here, 'xxx' represents one or multiple allowed actions. The actions in Step 1 are of the highest execution priority, while those in Step 2 and Step 3 are of medium and lowest execution priority.

4.The meta plan should reasonably arrange the division of action between Alice and Bob in order to achieve the goal as soon as possible.

Here is an example for you:

{Goal: [Transport 3 pens, 1 lighter, and 3 iPods to the bed.]
Meta plan: [Step 1: Alice explores the current room. Bob explores the current room.
Step 2: If any target objects are found, Alice and Bob  go grasp objects, put them into containers, and transport them to the bed.
Step 3: Alice goes to one of the remaining rooms and explores it. Bob goes to one of the remaining rooms and explores it]}

Goal: $GOAL$

Given the above goal, think step by step, and generate the meta plan:

---

Figure 9: Prompts for LLM in generating meta plan.

I am Alice. My teammate Bob and I want to transport as many target objects as possible to the bed with the help of containers within 3000 steps. I can hold two things at a time, and they can be objects or containers. I can grasp containers and put objects into them to hold more objects at a time.

Assume that you are an excellent leader for coordinating the task. Given our shared goal, meta plan, dialogue history, latest progress, and my previous actions,, please help me generate a message sent to Bob, in order to share my progress and inquire the opinion of Bob about the meta plan. Note that a container can contain three objects, and will be lost once transported to the bed. I can only put objects into the container I hold after grasping it. All objects are denoted as <name> (id), such as <table> (712). Actions take several steps to finish. It may be costly to go to another room or transport to the bed, use these actions sparingly.

The generated message should strictly meet following requirements:

The message has to be concise, reliable, and helpful for assisting Bob and me to make an efficient and consistent action plan, and transport as many objects to the bed as possible. Don't generate repetitive messages.

Here is an example of generated massage for you:

Example:
{Message: Hi Bob, I am exploring the <living room> (2000) and I found <apple> (1242543) there. I made an action plan to guide us to complete the task efficiently. Do you have any suggestions for the action plan according to your latest progress?}

Following are provided information for you:

Goal: \$GOAL\$

Previous meta plan: \$PREVIOUS\_PLAN\$

Dialogue history: \$DIALOGUE\_HISTORY\$

Progress: \$PROGRESS\$

Teammate progress: \$OPP\_PROGRESS\$

Figure 10: Prompts for LLM in the communication module of the mentor agent, e.g., Alice.

I am Bob. My teammate Alice and I want to transport as many target objects as possible to the bed with the help of containers within 3000 steps. I can hold two things at a time, and they can be objects or containers. I can grasp containers and put objects into them to hold more objects at a time.

Assume that you are an excellent leader for coordinating the task. Given our shared goal, action plan, dialogue history, progress, and my previous actions, please help me analyze the feasibility of the action plan proposed by Alice and generate a message to send to Alice. Note that a container can contain three objects, and will be lost once transported to the bed. I can only put objects into the container I hold after grasping it. All objects are denoted as <name> (id), such as <table> (712). Actions take several steps to finish. It may be costly to go to another room or transport to the bed, use these actions sparingly.

The generated meta plan should strictly meet following requirements:

1.The message has to be concise, reliable, and helpful for assisting Bob and me to make an efficient and consistent action plan, and transport as many objects to the bed as possible. Don't generate repetitive messages.

2.The message must strictly be in the following format: {Main message: a detailed opinion and suggestions of Bob about the action plan. Satisfaction level: Yes or No, decide whether you are satisfied with the current action plan.} You should consider the progress of both Alice and Bob in determining the satisfaction level and providing suggestions for the action plan.

3.If you are not satisfied with the current action plan, please point out the reason and your suggestion on how to modify the action plan in the message. You can suggest the next action for Alice in the message to achieve the goal as soon as possible.

Here is an example of generated massage for you:

Example:
Message: { Main message: Hi Alice, your proposed action plan looks great. However, considering that I found an <wood\_basket> (1870213) and <plastic\_basket>(1843721) in the <office>(2000), I think I should grasp <wood\_basket> (1870213), I suggest you to go to <office>(2000) to grasp <plastic\_basket>(1843721). Please update the meta plan. Satisfaction level: No}

Following are provided information for you:

Goal: \$GOAL\$

Previous meta plan: \$PREVIOUS\_PLAN\$

Dialogue history: \$DIALOGUE\_HISTORY\$

Progress: \$PROGRESS\$

Teammate progress: \$OPP\_PROGRESS\$

Figure 11: Prompts for LLM in the communication module of the teammate agent, e.g., Bob.

I am Alice. My teammate Bob and I want to transport as many target objects as possible to the bed with the help of containers within 3000 steps. I can hold two things at a time, and they can be objects or containers. I can grasp containers and put objects into them to hold more objects at a time.

Assume that you are an expert plan outline designer. Given our shared goal, previous meta plan, dialogue history, latest progress, please help me refine the meta plan into a more comprehensive and efficient plan for Bob and me, in order to achieve the goal collaboratively as soon as possible. Note that a container can contain three objects, and will be lost once transported to the bed. I can only put objects into the container I hold after grasping it. All objects are denoted as <name> (id), such as <table> (712). Actions take several steps to finish. It may be costly to go to another room or transport to the bed, use these actions sparingly.

The generated meta plan should strictly meet following requirements:

1.The meta plan should be brief, reliable, authentic, and consistent with the latest progress of Alice and Bob. Don't make random and meaningless plans.

2.There are 5 allowed actions you can use to construct the meta plan. 1) 'go to': move to a specified room. 2) 'explore': explore a room for underlying target objects. 3) ' go grasp': go to grasp a specified target object. 4) 'put': Place an object into a specified container. 5) 'transport': Transport holding objects or containers to the bed and drop them on the bed.

3.The meta plan must be structured strictly in a three-step format: {Action Plan: Step 1: Alice xxx, Bob xxx; Step 2: Alice xxx, Bob xxx; Step 3: Alice xxx, Bob xxx}. Here, 'xxx' represents one or multiple allowed actions. The actions in Step 1 are of the highest priority, while those in Step 2 and Step 3 are of medium and lowest priority, respectively.

4.The meta plan should reasonably arrange the division of action between Alice and Bob in order to achieve the goal as soon as possible.

Following are provided information for you:

Goal: \$GOAL\$

Previous meta plan: \$PREVIOUS\_PLAN\$

Dialogue history: \$DIALOGUE\_HISTORY\$

Progress: \$PROGRESS\$

Teammate progress: \$OPP\_PROGRESS\$

Figure 12: Prompts for LLM in cooperative planning module to generate progress-adaptive meta plan.

I am \$AGENT\_NAME\$. My teammate \$OPP\_NAME\$ and I want to transport as many target objects as possible to the bed with the help of containers within 3000 steps. I can hold two things at a time, and they can be objects or containers. I can grasp containers and put objects into them to hold more objects at a time.

Assume that you are an expert decision maker. Given our shared goal, action plan, my progress, and previous actions, please help me choose the best available action to achieve the goal as soon as possible. Note that a container can contain three objects, and will be lost once transported to the bed. I can only put objects into the container I hold after grasping it. All objects are denoted as <name> (id), such as <table> (712). Actions take several steps to finish. It may be costly to go to another room or transport to the bed, use these actions sparingly.

Goal: \$GOAL\$

Meta plan: \$META\_PLAN\$

Dialogue history: \$DIALOGUE\_HISTORY\$

Progress: \$PROGRESS\$

Previous action: \$PREVIOUS\_ACTIONS\$

Action list: \$ACTION\_LIST\$

Figure 13: Prompts for LLM in the plan praising module.

