# OpenReview forum: "CaPo: Cooperative Plan Optimization for Efficient Embodied Multi-Agent Cooperation"
_ICLR.cc/2025/Conference — ICLR 2025 Poster_

### Official Review · Reviewer_Z2bg · 2024-11-03

**Soundness:** 4
**Presentation:** 4
**Contribution:** 3
**Rating:** 8
**Confidence:** 4

**Summary:**

This paper introduces a method called CaPo, which uses LLM agents having multi-turn conversations with each other to create meta-plans (a collection of subtasks for solving the overarching problem that are assigned to each agent). Not only does it precompute these tasks, it dynamically adjusts them over time as agents perceive new information or interact with objects in ways that impact the preexisting plan. By testing on realistic simulators emulating embodied household tasks, the authors show the efficacy of the approach compared to conventional LLM approaches and classical symbolic planners.

**Strengths:**

The authors did a nice job ablating the different modular components of their agent architecture. Oftentimes in the LLM agent literature plans are either made in a single-turn or an agent just refines what to do in its own mind, but introducing multi-turn group communication seems like an effective strategy if one has enough of a communication budget. The architecture itself seems highly reproducible, the benchmarks are challenging, and the impact of which LLM underlies communication is thoughtfully examined.

**Weaknesses:**

The experiments could benefit more from being evaluated on real humans collaborating with LLM agents on tasks such as Watch and Help. If the ultimate goal of this line of work is to build embodied agents that help humans, understanding how real humans complete the task differently with different agents is key. Extra evaluations could show the same efficiency metrics but also metrics indicating which agents human preferred playing with. If CaPo is very verbose when making plans, humans may not prefer interacting with it despite the efficiency gains compared to a method like CoELA. It would be interesting to dissect this.

Moreover, an analysis of the runtime cost of CaPo would be nice to see. If a multi-round discussion is needed to reformulate a plan every time new information is discovered, how long does this take? How long can prompts end up being for the LLMs (how large does the memory grow as well) and does the duration of an episode/memory size impact planner quality?

**Questions:**

What is the standard error for the metrics shown in Table 1?

What is the runtime of this method at each turn? Can it be run in real-time with humans and how does planner quality change as a function of time?

Nice to have - what are results from playing this method with humans compared to other baselines? What do humans prefer?

---

> ### Author Response · Authors · 2024-11-21
> **Response to Reviewer Z2bg**
>
> Thank you for the insightful and positive comments! In the following, we provide our point-by-point response and hope our response helps address your concerns. We also look forward to the subsequent discussion which may further help to solve the current issues.
>
> ***Q1.The experiments could benefit more from being evaluated on real humans collaborating with LLM agents on tasks such as Watch and Help. Extra evaluations could show the same efficiency metrics but also metrics indicating which agents humans preferred playing with. If CaPo is very verbose when making plans, humans may not prefer interacting with it despite the efficiency gains compared to a method like CoELA. It would be interesting to dissect this.
> Nice to have - what are results from playing this method with humans compared to other baselines? What do humans prefer?***
>
> To evaluate human-agent cooperation, we follow the methodology of CoELA and conduct experiments on the C-WAH tasks, where the agent Bob (meta-plan evaluator) is controlled by real human participants. We adopt **average steps** as an efficiency metric and also ask three participants to rate their teammate agent (i.e., CaPo or CoELA) on a 7-point Likert Scale based on three criteria: **communication effectiveness, helpfulness, and trust**. The following results and discussion have been included in Appendix A7.
>
> | **Method**       | **Average Steps** | **Comm Effectiveness** | **Helpfulness** | **Trust**         |
> |-------------------|-------------------|-------------------------|-----------------|-------------------|
> | **CoELA**         | 47 ± 2           | 5.8 ± 0.3 / 7          | 5.5 ± 0.4 / 7   | 6.0 ± 0.1 / 7     |
> | **CaPo (ours)**   | 44 ± 1           | 6.3 ± 0.1 / 7          | 6.1 ± 0.3 / 7   | 6.3 ± 0.2 / 7     |
>
> Based on above table, we have following findings:
>
> - **Efficiency (Average steps).** Humans collaborating with CaPo generally achieve higher efficiency, completing tasks with fewer steps, i.e., 44 steps. This is because both agents and humans can avoid redundant actions through the guidance of the proposed meta-plan.
>
> - **Human Preferences**
>    - **Communication effectiveness** represents the effectiveness of communication between agent and human participants, e.g., whether the agent understands the message and provides useful information.  In general, **CaPo demonstrates clearer and more reliable communication than CoELA (6.3 vs. 5.8)**. This improvement is attributed to CaPo’s multi-round discussion mechanism, which creates a long-term, coherent meta-plan, reducing the need for frequent communication. In contrast, in CoELA, agents and humans have to exchange information more frequently, due to the lack of  a coherent plan. Finally, agents didn’t show verbose behavior when making plans (as shown in Figure 3), as we constrained the LLM to generate concise plans and messages in prompting.
>    - **Helpfulness** indicates whether the agent helps human participants complete the task faster. By multi-round discussion and creating coherent meta-plans, **CaPo could help both agent and human participants to reduce redundant actions in the embodied environments**, e.g., aimless exploration.
>    - **Trust** measures how human participants trust their agent teammate. **Humans show a stronger preference for trusting CaPo (6.3 vs. 6.0) during the cooperation**. This is because **CaPo involves humans in the planning process**, seeking their input during multi-round discussions to collaboratively create the meta-plan. In contrast, CoELA agents act more independently and sometimes fail to respond to human messages, which weakens trust.
>
> Through human-agent cooperation experiments, we demonstrate that CaPo enhances communication clarity, actively supports human decision-making, and fosters trust by engaging humans in the planning process. These advantages position CaPo as a more favorable teammate for human collaborators, underscoring its potential for real-world cooperative applications.

---

> > ### Author Response · Authors · 2024-11-21
> > **Response to Reviewer Z2bg**
> >
> > ***Q2. Moreover, an analysis of the runtime cost of CaPo would be nice to see. If a multi-round discussion is needed to reformulate a plan every time new information is discovered, how long does this take? How long can prompts end up being for the LLMs (how large does the memory grow as well) and does the duration of an episode/memory size impact planner quality?
> > What is the runtime of this method at each turn? Can it be run in real-time with humans and how does planner quality change as a function of time?***
> >
> > To address the runtime concerns, we conducted experiments on the food tasks of TDW-MAT under oracle perception and GPT-3.5 settings. Below are the key findings:
> >
> > | **Method** | **Average running time cost per task** | **Average time cost per meta-plan update** | **Number of meta-plan updates** | **Average prompt length** |
> > |:----------:|:-------------------------------------:|:-----------------------------------------:|:-------------------------------:|:-------------------------:|
> > | **CaPo**   | 26.2 min                              | 20.4 seconds                              | 14 times                        | 5 ~ 6K characters         |
> >
> >
> >
> > - **Runtime per task and turn**:
> >     - Each task takes an average of 26.2 minutes to complete, with approximately 14 meta-plan updates per task.
> >     - Each meta-plan update involves up to 3 discussion rounds (resulting in 6 messages), taking an average of 20.4 seconds per meta-plan update.
> >     - Generating a single message or plan takes approximately 3.4 seconds, which is acceptable for real-time human interaction.
> >
> > - **Prompt length and Memory**:
> >   - Prompt lengths can increase as the task progresses, as they include dialogue history and task progress descriptions.
> >   - To mitigate excessive memory growth, we follow CoELA’s approach by using a fixed memory module size, storing only the most recent 10 messages or progress in the memory.
> >   - This keeps the average prompt length at 5–6K characters, which is within the operational limits of GPT-3.5 and comparable to CoELA.
> >
> > - **Impact of Episode Duration and Memory Size on Planner Quality**:
> >     - Due to the fixed memory size, we did not observe any significant impact of episode duration or memory size on planner quality.
> >     - The planner quality is instead reflected in task efficiency, as measured by transport rate. Figure 5 shows that CaPo consistently achieves higher transport rates under varying time budgets, demonstrating that the meta-plan’s quality and effectiveness are maintained over time.
> >
> > - **Real-Time Feasibility**:
> > CaPo is designed to operate efficiently in real-time with humans, as the runtime of each turn (20.4 seconds for a full meta-plan update) is well within the range for practical interactions. Additionally, the constrained prompt lengths and fixed memory size ensure computational feasibility and consistency during extended tasks.
> >
> > ***Q3. What is the standard error for the metrics shown in Table 1?***
> > - Since the API of  GPT 3.5/4 is expensive, we follow prior work like CoELA to run our experiments one time and compare with these priors.
> >
> > - To further address your concerns, we run our experiments multiple (3) times, and find our method is robust. For example, under the oracle perception setting on the TDW-MAT task, our CaPo with GPT-3.5 achieves a transport rate of 0.83 and enjoys a very small variance of 0.012.  These results demonstrate the stability and reproducibility of our method. We have included these experiments into Table 6 of the Appendix.

---

> > > ### Comment · Reviewer_Z2bg · 2024-11-24
> > >
> > > Thank you for your response and additional analysis. This was very insightful and a useful statistic to keep in the paper, or at least the appendix, for further work to evaluate the efficiency of their agents, both in terms of number of samples and planning steps but also in terms of a much more significant real world metric of time taken to complete a task or make a decision. One might imagine that an agent could complete a task in less "steps," but if it takes longer to complete the task in terms of seconds this might lead to a false sense of efficiency. Luckily, it seems like your approach does better on both fronts, and I'm happy to increase my score!

---

### Official Review · Reviewer_kpSn · 2024-11-04

**Soundness:** 3
**Presentation:** 3
**Contribution:** 2
**Rating:** 5
**Confidence:** 3

**Summary:**

This paper introduces the Cooperative Plan Optimization (CaPo) framework, which aims to enhance the long-term planning cooperation efficiency among Large Language Model (LLM)-based embodied agents. CaPo includes two-phase: meta-plan generation and progress-adaptive meta-plan and execution. Experimental results on the TDW-MAT and C-WAH tasks demonstrate CaPo's significant improvement over state-of-the-art methods.

**Strengths:**

The paper introduces a novel two-phase planning approach, Cooperative Plan Optimization (CaPo), which addresses the limitations of single-step planning in CoELA. This approach demonstrates its effectiveness through experiments on the TDW-MAT and C-WAH environments.

**Weaknesses:**

The main content of the article appears to be an extension of CoELA[1], enhancing its single-step planning approach by incorporating multi-turn discussions after the execution phase. Meanwhile, several works, for example, [2,3] have already demonstrated multiple rounds of discussions in various contexts, so the novelty and significance of this article's contribution seem somewhat lacking.

Additionally, I confess that I am not familiar with ThreeDworld Multi-Agent Transport (TDW-MAT) task, and the Communicative Watch-And-Help (C-WAH) task, so I am not quite clear on how to assess their complexity. However, for works [4,5,6] based on Minecraft, which typically evaluates dozens or even hundreds of tasks. I am cautiously skeptical about whether considering only 6 or 10 target tasks in food and stuff settings is sufficient to thoroughly validate the capabilities of the proposed framework. How the complexity of TDW-MAT and C-WAH tasks compares to Minecraft tasks? Whether you have plans to evaluate on a larger set of tasks in future work?

[1] Zhang, H., Du, W., Shan, J., Zhou, Q., Du, Y., Tenenbaum, J.B., Shu, T. and Gan, C., 2023. Building cooperative embodied agents modularly with large language models. arXiv preprint arXiv:2307.02485.
[2] Du, Y., Li, S., Torralba, A., Tenenbaum, J.B. and Mordatch, I., 2023. Improving factuality and reasoning in language models through multiagent debate. arXiv preprint arXiv:2305.14325.
[3] Wang, Z., Mao, S., Wu, W., Ge, T., Wei, F. and Ji, H., 2023. Unleashing the emergent cognitive synergy in large language models: A task-solving agent through multi-persona self-collaboration. arXiv preprint arXiv:2307.05300.
[4] Wang, Z., Cai, S., Chen, G., Liu, A., Ma, X. and Liang, Y., 2023. Describe, explain, plan and select: Interactive planning with large language models enables open-world multi-task agents. arXiv preprint arXiv:2302.01560.
[5] Wang, G., Xie, Y., Jiang, Y., Mandlekar, A., Xiao, C., Zhu, Y., Fan, L. and Anandkumar, A., 2023. Voyager: An open-ended embodied agent with large language models. arXiv preprint arXiv:2305.16291.
[6] Cai, S., Zhang, B., Wang, Z., Ma, X., Liu, A. and Liang, Y., 2023. Groot: Learning to follow instructions by watching gameplay videos. arXiv preprint arXiv:2310.08235.

**Questions:**

In the ablation study, the paper mentions the impact of the number of agents, but the game appears to only contain two players, Alice and Bod. I'm confused about the number of agents. Please clarify how you tested different numbers of agents if the main experiments only used two.

---

> ### Author Response · Authors · 2024-11-20
> **Response to Reviewer kpSn**
>
> Thank you for the insightful and valuable comments! In the following, we provide our point-by-point response and hope our response helps address your concerns. We also look forward to the subsequent discussion which may further help to solve the current issues.
>
> ***Q1: The main content of the article appears to be an extension of CoELA, enhancing its single-step planning approach by incorporating multi-turn discussions after the execution phase.***
>
> - We respectfully disagree that the contributions are just an extension of CoELA compared to the original CoELA framework.  **We introduce two key innovative strategies**, that are not in the original framework, and that significantly improve the cooperation efficiency of embodied agents in the experiments:
>   - **Meta-plan generation mechanism** that provides long-term strategic guidance, enabling agents to form coherent, goal-aligned plans that enhance coordination efficiency in decentralized settings. By contrast, CoELA relies on ad-hoc or reactive adjustments.This meta-plan mechanism ensures consistent and structured decision-making, crucial for complex multi-agent tasks.
>
>   - **Progress-adaptive meta-plan and execution strategy** that dynamically adjusts and refines the meta-plan based on real-time progress, ensuring adaptability and responsiveness in unpredictable environments. This adaptiveness addresses the limitations of static or pre-defined strategies, particularly in scenarios with unexpected challenges.
>
> - These contributions are **both novel and impactful**, advancing the field of LLM-driven embodied agent cooperation by introducing a **dynamic escalation mechanism**. Unlike traditional methods such as CoELA, this mechanism enables agents to actively detect and discuss potential errors or inconsistencies, particularly in decentralized settings with partial observations. Such environments often lack shared global information, leading to low cooperation efficiency, as indicated in Table 3. By introducing active intervention, our strategies prevent error propagation and significantly enhance system robustness and efficiency, as demonstrated in Tables 1 and 2.
>
>
> ***Q2: [2,3] have already demonstrated multiple rounds of discussions in various contexts, so the novelty and significance of this article's contribution seem somewhat lacking.***
>
> - Thank you for bringing up prior works [2, 3] on multi-round discussions, which have indeed explored this concept in different contexts. Specifically, [2] focuses on enhancing the mathematical and strategic reasoning capabilities of large language models (LLMs) through multi-round debates between distinct LLM instances. On the other hand, [3] introduces a solo performance prompting strategy, transforming a single LLM into a cognitive synergist by engaging in multi-turn self-collaboration using multiple personas.
>
> - Our work, however, fundamentally differs from these approaches in two key aspects:
>     - **Distinct Objectives:** While [2] and [3] investigate multi-round discussions to improve factuality, reasoning ([2]), or cognitive synergy ([3]) of LLMs, our work is specifically designed to enhance cooperation efficiency among embodied agents in decentralized settings, where agents operate with partial observations. This shift in focus addresses a unique and practical challenge not considered in the aforementioned studies.
>
>     - **Innovative Strategies:** We introduce two novel strategies tailored for embodied agent systems: the Meta-Plan Generation Mechanism and the Progress-Adaptive Meta-Plan and Execution Strategy. These strategies are explicitly designed to significantly improve coordination and efficiency in decentralized multi-agent environments, addressing complexities that are beyond the scope of [2] and [3].
>
> - By focusing on decentralized embodied agent cooperation and offering innovative, application-specific strategies, our work addresses a unique problem space and provides a tailored solution distinct from prior methods.

---

> > ### Author Response · Authors · 2024-11-20
> > **Response to Reviewer kpSn**
> >
> > ***Q3: The main experiments are conducted on TDW-MAT and C-WAH tasks, is it sufficient to validate the effectiveness of the proposed model? How does the complexity of TDW-MAT and C-WAH tasks compare to Minecraft tasks? Do you have plans to evaluate a larger set of tasks in the future work?***
> >
> > - **Regarding task complexity**, Minecraft tasks primarily emphasize open-ended exploration and instruction-following in static, predictable environments. They are geared towards multi-task learning and long-term exploration strategies. In contrast, TDW-MAT and C-WAH tasks highlight the challenges of dynamic, decentralized environments with partial observations. These settings require robust decentralized coordination and real-time communication, making them particularly relevant for studying embodied multi-agent cooperation. Given this focus, we do not evaluate our model on environments like Minecraft, as it falls outside the scope of our study.
> >
> > - For fairness, we compare our method with other baselines on the **TDW-MAT and C-WAH tasks, as these are widely recognized benchmarks in prior work (Guo et al., 2024; Zhang et al., 2023b)** for evaluating embodied multi-agent cooperation systems.
> >
> > - Our focus in this work is on the unique challenges posed by TDW-MAT and C-WAH. These tasks are specifically designed to test scenarios involving **partial observations, decentralized agent control, and complex agent coordination**—key aspects that differ significantly from the objectives of more open-ended and exploratory tasks like those in Minecraft.
> >
> > - To further address your concerns, **we extended our evaluation to additional environments, such as RoCobench (Mandi et al., 2023)**. We selected three tasks from RoCobench that involve asymmetric observations, which align more closely with the decentralized, partial-observation settings of our study. The results, presented above, show that CaPo achieves a higher success rate with fixed steps and requires fewer steps to complete tasks, demonstrating the effectiveness and generalization of our model for embodied multi-agent cooperation.
> >
> > | GPT-4-driven Agent |       Sweep Floor       |                      |      Pack Grocery      |                      |        Move Rope       |                      |
> > |:-------------------:|:----------------------:|:--------------------:|:---------------------:|:--------------------:|:----------------------:|:--------------------:|
> > |                     |    Success rate        |        Step          |     Success rate       |        Step          |     Success rate        |        Step          |
> > |       **RoCo**      |       0.95±0.05        |         7.1          |        0.44±0.06       |         9.9          |        0.65±0.11        |         2.5          |
> > |    **CaPo (ours)**  |       0.98±0.13        |         6.5          |        0.63±0.11       |         7.6          |        0.74±0.17        |         2.2          |
> >
> > - Looking ahead, **we are committed to expanding our evaluations post-rebuttal when more time allows**. Specifically, we plan to benchmark our method on larger-scale datasets, including recently released embodied multi-agent benchmarks [1]. This will further validate the scalability and applicability of our approach across diverse environments."
> >
> > [1] Chang, Matthew, et al. "PARTNR: A Benchmark for Planning and Reasoning in Embodied Multi-agent Tasks." arXiv preprint arXiv:2411.00081 (2024).
> >
> > ***Q4: How do you test different numbers of agents if the main experiments only used?***
> >
> > - To ensure a fair comparison with existing baselines, our main experiments focus on the two-agent setup. However, to explore the scalability and robustness of our approach, we extend our evaluation to multi-agent scenarios (more than two agents) in the ablation study, specifically using the C-WAH tasks.
> >
> > - In these ablation experiments, one agent is designated as the meta-plan designer, while the remaining agents evaluate the proposed meta-plan and provide feedback. The discussion process concludes either when all agents reach a consensus on the updated meta-plan or when the communication budget (three rounds) is exhausted. After this, all agents collaboratively execute the agreed-upon meta-plan until the next discussion is triggered or the task concludes.
> >
> > - This multi-agent ablation study allows us to assess how well our approach scales to more complex, decentralized scenarios with multiple agents. The results demonstrate the robustness of our method in maintaining cooperation efficiency and task performance, even as the number of agents increases.

---

### Official Review · Reviewer_zu5a · 2024-11-04

**Soundness:** 2
**Presentation:** 3
**Contribution:** 1
**Rating:** 5
**Confidence:** 5

**Summary:**

The paper introduces Cooperative Plan Optimization (CaPo) to improve cooperation among LLM-based embodied agents. Existing methods lack strategic planning, leading to inefficiency and failures in complex cooperative tasks. CaPo enhances cooperation through two phases: meta-plan generation and progress-adaptive execution. In the first phase, agents collaboratively analyze the task and create a strategic plan with detailed steps. In the second phase, agents execute and adjust the plan dynamically based on real-time progress, supported by ongoing multi-turn discussions. This progress-based adaptation prevents redundant actions and boosts overall task efficiency. Empirical experiments on the ThreeDworld Multi-Agent Transport and Communicative Watch-And-Help tasks demonstrate CaPo’s superiority in task completion and efficiency.

**Strengths:**

1. The paper is well-written and easy to follow. The figures and illustrative examples help demonstrate the effectiveness of the proposed progress-adaptive meta-plan.
2. The related work section is sufficiently comprehensive, providing a clear context for the study.
3. Complex ablation studies on each module of CaPo, the effect of agent number, and convergence analysis.

**Weaknesses:**

1. This work appears to be a straightforward extension of CoELA, with minimal innovation beyond basic prompt engineering for the meta-plan module. The problem setting and agent framework largely mirror those of CoELA. For instance, as noted in Section 4.1, the CaPo agent comprises seven modules: 1) perception, 2) memory, 3) communication, 4) cooperative planning, 5) progress-adaptive planning, 6) plan parsing, and 7) execution. The CoELA agent already includes five of these, lacking only the cooperative planning and progress-adaptive planning modules, while these 2 modules can be easily implemented by prompting LLM.
2. The authors limited their experiments to the TDW-MAT and C-WAH testbeds, which are already used in CoELA. To demonstrate the broader effectiveness of their method, it would be beneficial to include experiments in new and varied environments.
3. The process of updating the meta-plan relies on broadcasting communication among agents, which incurs significant communication costs.

**Questions:**

1. Can the authors elaborate further on their contributions compared to CoELA? The current presentation does not sufficiently highlight the novelty of this work.
2. Would the authors consider conducting additional experiments in different environments to better demonstrate the generalizability and effectiveness of their method?
3. Is it possible to reduce the communication cost by replacing the broadcasting communication mechanism with a different network design?
4. In Figure 8, the authors claim that consensus is typically reached within 3 rounds. However, over 20% of trials still fail to reach consensus after 3 rounds. Is setting the discussion budget to 3 rounds a reasonable choice given this observation?

---

> ### Author Response · Authors · 2024-11-20
> **Response to Reviewer zu5a**
>
> Thank you for the insightful and valuable comments! In the following, we provide our point-by-point response and hope our response helps address your concerns. We also look forward to the subsequent discussion which may further help to solve the current issues.
>
> ***Q1:  Can the author further illustrate their contribution compared to CoELA? CaPo reused some modules from CoELA, and 2 proposed modules can be easily implemented by prompting LLM?***
>
> - We respectfully disagree that the contributions are limited compared to the original CoELA framework.  **We introduce two key innovative strategies**, that are not in the original framework, and that significantly improve the cooperation efficiency of embodied agents in the experiments:
>   - **Meta-plan generation mechanism** that provides long-term strategic guidance, enabling agents to form coherent, goal-aligned plans that enhance coordination efficiency in decentralized settings. By contrast, CoELA relies on ad-hoc or reactive adjustments.This meta-plan mechanism ensures consistent and structured decision-making, crucial for complex multi-agent tasks.
>   - **Progress-adaptive meta-plan and execution strategy** that dynamically adjusts and refines the meta-plan based on real-time progress, ensuring adaptability and responsiveness in unpredictable environments. This adaptiveness addresses the limitations of static or pre-defined strategies, particularly in scenarios with unexpected challenges.
>
> - These contributions are **both novel and impactful**, advancing the field of LLM-driven embodied agent cooperation by introducing a **dynamic escalation mechanism**. Unlike traditional methods such as CoELA, this mechanism enables agents to actively detect and discuss potential errors or inconsistencies, particularly in decentralized settings with partial observations. Such environments often lack shared global information, leading to low cooperation efficiency, as indicated in Table 3. By introducing active intervention, our strategies prevent error propagation and significantly enhance system robustness and efficiency, as demonstrated in Tables 1 and 2.
>
> - Moreover, **for the reused modules**, i.e., perception,  memory, communication, and execution modules, **we have moved them into the preliminary section**, and only introduced the improvements on these modules in our method section. For example, comparing vanilla memory modules, we introduce plan related information like meta plan and task progress into the memory module for retrieval to discuss new meta plan or adaptation. Please refer to these changes in our new submission which will be polished better later. This will help readers clearly understand our main innovative contribution. We also emphasize that, throughout the entire initial submission, we did not claim these reused modules from CoELA as our contribution.
>
> - Finally, while our strategies are implemented through LLM prompting—a standard technique in LLM-based embodied agent methods such as CoELA, RoCo (Mandi et al., 2023), and ProAgent (Zhang et al., 2023b)—**our novelty lies in the design and application of these strategies (meta-plan generation and its progressive adaptation) and their profound impact on agent cooperation, rather than the use of prompting itself**. That is, in this work, prompting is only the implementation tool of our novel strategies instead of the key contribution.

---

> > ### Author Response · Authors · 2024-11-20
> > **Response to Reviewer zu5a**
> >
> > ***Q2: Current experiments are limited to the TDW-MAT and C-WAH testbeds, would the authors consider conducting additional experiments in different environments to better demonstrate the generalizability and effectiveness of their method?***
> >
> > - **For fairness, we compare our method with baselines on the TDW-MAT and C-WAH tasks**, since these two tasks are well-recognized in prior work (Guo et al., 2024, Zhang et al., 2023b) as effective benchmarks for evaluating embodied multi-agent cooperation systems.
> >
> > - **As per your suggestions, we further test the generalization and effectiveness of our model  on the RoCk Bench (Mandi et al., 2023)**, which is a multi-robot collaboration environment. We choose three tasks exhibiting asymmetric observation, which are more relevant to  decentralized, partial-observation settings. We only compare CaPo with RoCo, since it often achieves better performance than CoELA and ProAgent, and the rebuttal period is limited. The results below show that CaPo achieves better performance under two settings: a) it has a higher success rate when there are restrictions on maximum steps (10 steps); 2) CaPo requires fewer steps to complete the task when there is no step restriction. This further highlights the effectiveness and generalization of our model on embodied multi-agent cooperation.
> >
> > | GPT-4-driven Agent |       Sweep Floor       |                      |      Pack Grocery      |                      |        Move Rope       |                      |
> > |:-------------------:|:----------------------:|:--------------------:|:---------------------:|:--------------------:|:----------------------:|:--------------------:|
> > |                     |    Success rate     |        Step          |     Success rate       |        Step          |     Success rate        |        Step          |
> > |       **RoCo**      |       0.95±0.05        |         7.1          |        0.44±0.06       |         9.9          |        0.65±0.11        |         2.5          |
> > |    **CaPo (ours)**  |       0.98±0.13        |         6.5          |        0.63±0.11       |         7.6          |        0.74±0.17        |         2.2          |
> >
> > ***Q3: The update of meta plan may introduce significant communication cost, is it possible to reduce the communication cost by replacing the broadcasting communication mechanism with a different network design***
> >
> > For communication costs, we discuss them from several aspects.
> >
> > - Firstly, the task cost  mainly includes agent moving cost and communication cost. By comparison,  **moving cost is often much higher than communication cost**, since current robots lack the ability of humans, and their movement is costly especially in complex and large-scale environments like urban areas; while the latter only needs LLM’s inference, and massage sending and receiving, and is indeed much easier and efficient. For example, moving from one room to another can often require much more time than LLM’s inference.
> >
> > - Moreover, **to reduce communication cost, our CaPo adopts two strategies.** a) agent’s progress-adaptive planning module will evaluate whether meta plan is  suitable for current task progress: if yes, then all agents will continue their subtasks, and will not communicate, reducing communication costs; b) with the meta plan as a reference, a few round communication is sufficient to arrive at a new plan, since task progress is often smoothly achieved and thus adaptation the meta plan need slightly small changes. Indeed, we set the maximum round of communication as 3 for each discussion. So these strategies differ from CoELA which needs communication at each iteration, while ours does not.
> >
> > - **To further address your concerns, we report the total running time on the CWAH dataset**, including moving and communication time. For visual Object setting on CWAH,  CoLEA and our CaPo respectively complete all tasks in 56.7 and 62.5 minutes, while needing an average of 92 and 83 moving steps on each task. This shows CaPo’s balanced runtime efficiency without significant added delays.
> >
> > | Method       | Running Time Cost | Average Steps |
> > |:------------:|:-----------------:|:-------------:|
> > | CoELA        |     56.7 min      |      92       |
> > | CaPo (ours)  |     62.5 min      |      83       |
> >
> > - **Replacing the broadcasting mechanism with different network designs**, such as selective or hierarchical communication design, could potentially reduce communication costs, and is promising. For selective or hierarchical communication design, agents are allowed to discuss point-to-point instead of broadcasting, thereby getting rid of many unnecessary message sending and receiving, and improving communication efficiency. Due to the limited rebuttal period, we will leave this interesting direction as our future work.

---

> ### Author Response · Authors · 2024-11-20
> **Response to Reviewer zu5a**
>
> ***Q4:  Is setting the discussion budget to 3 rounds a reasonable choice given this observation?***
>
> **We set the discussion budget to 3 rounds to balance the consensus rate and overall efficiency.** As shown in the following table, with a 3-round limit, the failure rate decreases to 21.1%, and overall efficiency improves to 84%. Extending the discussion budget beyond 3 rounds offers only marginal benefits in reducing failed cases (e.g., a reduction to 17.8% with 4 rounds) while providing no additional efficiency gains, as efficiency stabilizes at 84%. These results demonstrate that a 3-round discussion budget is a reasonable choice, effectively balancing communication costs and task performance. We include these findings in the supplementary material for added clarity.
>
> | Maximal Round       |   1   |   2   |   3   |   4   |   5   |
> |:-------------------:|:-----:|:-----:|:-----:|:-----:|:-----:|
> | Rate of failed cases| 79.1% | 53.5% | 21.1% | 17.8% | 15.5% |
> | Transport rate      |  79%  |  82%  |  84%  |  84%  |  83%  |

---

> ### Comment · Reviewer_zu5a · 2024-11-25
> **Remaining Concerns**
>
> I thank the authors for their response and for addressing my questions. I appreciate the additional experiments on RoCoBench, which provide further evidence of the proposed framework's effectiveness. However, several fundamental weaknesses in this work remain unresolved.
>
> First, the novelty of the proposed framework remains questionable. As noted in my initial review and echoed by Reviewers 8MR1 and kpSn, this work appears to be a relatively straightforward extension of CoELA. The response simply restates the introduction of Meta-plan generation and Progress-adaptive meta-plans as key contributions, which primarily rely on trivial prompt engineering.  Furthermore, the performance improvements over baselines appear to hinge on the communication cost introduced by the "central agent." This represents a trade-off rather than a clear advancement, and the authors have not provided a thorough evaluation of the communication cost. I will elaborate on this issue in the next section.
>
> Secondly, the evaluation of communication cost remains unclear. As Reviewer 8MR1 noted, "the running time includes both the model's inference time and the simulator's simulation and render speed, which does not seem a good metric to evaluate the efficiency of the method." While the authors provided updated results in their second-round response to Reviewer 8MR1, the evaluation still lacks clarity and rigor in assessing efficiency. The authors assert that 500 characters correspond to 1 frame, a setting adopted from CoELA. However, I find this conversion problematic, as communication cost cannot be directly equated to step cost in this manner. The actual cost depends heavily on the underlying communication network and hardware. A more appropriate and widely accepted metric would be the communication bandwidth or overhead during execution, as demonstrated in prior works on multi-agent communication [1,2]. A fair comparison between CaPo, CoELA, and other baselines should be conducted using such metrics to provide a comprehensive evaluation.
>
> Given these unresolved issues, I am inclined to maintain my original rating.
>
> [1] Wang, Yuanfei, et al. "Tom2c: Target-oriented multi-agent communication and cooperation with theory of mind." ICLR 2022.
>
> [2] Ding, Ziluo, et al. "Learning individually inferred communication for multi-agent cooperation." NeurIPS 2020.

---

> > ### Author Response · Authors · 2024-11-25
> > **Second Response to Reviwer zu5a**
> >
> > Thank you for valuable feedbacks. We have carefully considered your new comments  and found that your main concerns are contribution and communication cost of our work. To this end, we provided detailed response below, hoping to address your concerns.
> >
> > ***Q1. Contribution Clarification***
> >
> > Thank you for your detailed feedback regarding our work. We appreciate the opportunity to clarify the core issues in embodied multi-agent cooperation tasks and our innovative solutions to address them:
> >
> > **Core Issues**: Embodied multi-agent cooperation involves several significant obstacles:
> >
> > - **Limited Observability**: Each individual agent has access to only partial information about the environment, posing substantial challenges to efficient collaboration and decision-making.
> >
> > - **Dynamic and Long-Horizon Tasks**: Embodied cooperation in long-horizon tasks usually require long-step execution (e.g., > 1k steps on TDW-MAT tasks) and real-time adjustments, which cannot be effectively addressed by single-step or static planning approaches.
> >
> > - **Effective Coordination**: Agents must avoid redundant actions while ensuring synchronized efforts, which is particularly challenging in decentralized multi-agent systems.
> >
> > While **existing methods**, such as COELA, provide partial solutions—for instance, in coordinating agents under limited observability—they are **constrained by their reliance on single-step planning**, which **lacks the flexibility needed to handle dynamic, multi-step tasks in complex environments**.
> >
> > **Innovative Solutions**: To tackle these obstacles, our work introduces a new approach that advances multi-agent cooperation through long-term and adaptive planning mechanisms:
> >
> > - **Dynamic Meta-Planning**: Our system employs a long-term meta-planning strategy to **provide agents with strategic guidance over multiple steps**, overcoming the limitations of static or short-term planning found in existing methods like COELA.
> >
> > - **Real-Time Adaptation**: We developed two novel modules—the Cooperative Planning Module and the Progress-Adaptive Planning Module—that enable continuous refinement of the meta-plan based on real-time task progress,  **dynamically enhancing the coordination of embodied agents**.
> >
> > We hope this clarification effectively highlights the unique contributions of our work and its impact on advancing embodied multi-agent cooperation. Thank you again for your constructive feedback.
> >
> > ***Q2. Communication cost***
> >
> > In our CaPo framework, we follow the same setting as CoELA for fair comparison, and **communication is considered costly, with 500 characters equating to a cost of 1 frame**.
> >
> > Accordingly, we provide a detailed analysis of the communication cost in the following table by collecting statistics on the TDW-MAT tasks.
> >
> > | Task     | Rounds of Meta-plan Updates | Character Cost per Meta-plan Updates | Frames Cost from Communication | Rounds of Discussion per Meta-plan Updates | Time Cost per Meta-plan Update | Time Cost per Task |
> > |-------------|-------------------------------|------------------------------|--------------------------------|---------------------------------------|-------------------------------|--------------------|
> > | **Food**  | 14                            | 1,757 chars                  | 56 frames                      | 2.4 rounds                           | 20.4 sec                      | 26.2 min          |
> > | **Stuff** | 17                            | 1,462 chars                  | 51 frames                      | 2.1 rounds                           | 22.7 sec                      | 25.6 min          |
> >
> > - Taking the food tasks as an example, our CaPo requires 14 meta-plan updates per task, with each update generating an average of 1,757 characters. Since 500 characters equate to 1 frame, this results in **a communication cost of 56 frames out of the total 3,000 frames for each task**. Similarly,  the communication cost for each task on Stuff is 51 frames.
> >
> > - As demonstrated by CoELA and noted by reviewer 8MR1, all steps (frames) consist of both moving steps and communication steps. Therefore, **the communication cost is inherently reflected in the final efficiency metric**, i.e., the transport rate. As shown in the table 1 and 2, our method exhibits significant efficiency improvement, highlighting a good tradeoff between communication cost and efficiency  improvement. We include these results and analysis in A.6 of supplementary materials.
> >
> > -  We appreciate the suggestion to use metrics such as communication bandwidth or overhead, we will cite these work [1,2],  and leave it for future work considering great workload and limited rebuttal time.
> >
> > We hope such analysis could address your concern about communication cost.

---

> ### Author Response · Authors · 2024-12-01
> **Thank you for your review and we are looking forward to your feedback!**
>
> Dear Reviewer zu5a,
>
> We sincerely appreciate your valuable comments. We have provided **a detailed response in the boxes above** to address your concerns. **Here is a summary**:
>
> - **Clarifying contribution**
>
>     - Both **reviewers 8MR1 and Z2bg acknowledged the contributions of this work** in advancing embodied multi-agent cooperation, particularly in **enhancing cooperation efficiency** through the **novel meta-plan mechanism and progress-adaptive meta-plan updating**.
>
>     - **To better highlight the contributions of this work, we have revised the manuscript**,  such as rewriting the introduction section and moving the introduction of the baseline CoELA to the preliminary section **for improved clarity**.
>
> - **Clarifying communication cost**
>
>   - We have provided a detailed evaluation of the communication cost in Appendix A6, demonstrating that our method achieves **a good balance between efficiency and communication cost**.
>
>   - Both reviewers 8MR1 and Z2bg acknowledged the **thoroughness and validity of our communication cost evaluation**.
>
>   - We have also **cited the papers you provided** and **will evaluate communication cost with bandwidth or overhead metrics** in future work, given the significant workload and limited rebuttal time.
>
> We sincerely wish that our response has addressed your concerns, and turned your assessment to the positive side. If you have any more questions, please feel free to let us know. We appreciate your time and constructive suggestions! Thank you!

---

> ### Author Response · Authors · 2024-12-03
> **Gentle reminder**
>
> Dear Reviewer zu5a,
>
> The discussion period closes in 1 day. We are trying best to **address all your concerns with new results, clarifications, and an updated manuscript**. Please let us know if you have any remaining concerns. We sincerely wish that our response has addressed your concerns, and turned your assessment to the positive side.
>
> Best,
>
> Authors

---

### Official Review · Reviewer_8MR1 · 2024-11-04

**Soundness:** 3
**Presentation:** 3
**Contribution:** 2
**Rating:** 6
**Confidence:** 4

**Summary:**

The paper introduced Cooperative Plan Optimization into the CoELA framework to improve embodied multi-agent cooperation. A new meta-plan mechanism ensures agents can form and follow a long-term strategic and coherent plan for efficient coordination. Multi-turn discussions following pre-defined templates among agents are triggered whenever new progress is made to adapt the meta plan to progress. The experiments on two embodied multi-agent cooperation benchmarks demonstrate the proposed method achieves better performance than sota.

**Strengths:**

- Introducing a meta-plan mechanism to ensure agents can form and follow a long-term strategic and coherent plan can improve the coordination's efficiency.

- The paper is clearly written and well-organized.

- The ablation study and qualitative analysis provides valuable insights.

**Weaknesses:**

- Since the paper reuses 1) a perception module, 2) a memory module, 3) a communication module, and 7) an execution module from the CoELA paper, it may be better to put the discussion of these modules into the preliminary sections rather than in the method section to make this paper's contribution more clear.

- The contribution is limited since the only improvement is achieved by introducing a pre-defined communication protocol by prompting into an existing CoELA framework.

- The paper uses the same formalization and experiments set up as prior work CoELA, which is not a bad thing. For the main results in Table 1, the column with RHP*, RHP, and LLAMA-2 CoELA is identical as reported in the CoELA paper but with no appropriate notation. In the meantime, only the CoELA baseline's results are changed to use GPT-3.5 backbones instead of the GPT-4 used in the original paper, which also causes a major performance drop (avg. 71 -> 52; 85->72) compared to the reported results in the original paper. Results with gpt-4 are reported as original in Table 2 and Appendix Table 5, where the performance gain is down to 4 points rather than 13 points. Why the results are organized as such needs further clarification.

- One of the interesting settings of the CoELA paper is that they formalized communication as a step that also comes with a cost so the agents need to balance the cost and gain of communication. They make the findings that it's not effective yet for LLM-based agents to communicate spontaneously, which may be part of the motivation for this paper. In this paper, however, it's not specifically discussed whether this costly communication is still employed, and if so, some analysis around the communication cost and corresponding efficiency achieved may provide valuable insights for the community.

- Is the progress-adaptive planning module always triggered whenever there is new progress? How many rounds of meta-plan updates are there on average and how many steps would be used to conduct this meta-plan discussion? Is the communication module also needed after the progress-adaptive planning module makes the update to share it with the others? What happens if two agents make new progress at the same time and all want to be the meta-plan designer? What will happen if agents can't reach a consensus? Will they just have no communication afterward or try to communicate again after how many steps or stuck at arguing every few steps?

- Minor typos:
  - L70: typo, 'sever' -> 'several'

**Questions:**

Please see the weaknesses above.

---

> ### Author Response · Authors · 2024-11-20
> **Response to Reviewer 8MR1**
>
> Thank you for the insightful and positive comments! In the following, we provide our point-by-point response and hope our response helps address your concerns. We also look forward to the subsequent discussion which may further help to solve the current issues.
>
> ***Q1: Better putting the discussion of the reused modules from CoELA into the preliminary sections.***
>
> Thank you for the suggestion! In revision, following your suggestion, we have moved this discussion into the preliminary section, and only introduced in the method section the improvements of our method on top of these modules. For example, on top vanilla memory modules, we introduce plan-related information like meta plan and task progress into the memory module for retrieval to discuss new meta plan or adaptation.
>
> ***Q2: The contribution is limited since the only improvement is achieved by introducing a pre-defined communication protocol by prompting into an existing CoELA framework.***
>
> - We respectfully disagree with the claim that the contributions are limited compared to the original CoELA framework.  **We introduce two key innovative strategies** that are not in the original framework and significantly improve the cooperation efficiency of embodied agents in the experiments:
>
>   - **Meta-plan generation mechanism** that provides long-term strategic guidance, enabling agents to form coherent, goal-aligned plans that enhance coordination efficiency in decentralized settings. By contrast, CoELA relies on ad-hoc or reactive adjustments.This meta-plan mechanism ensures consistent and structured decision-making, crucial for complex multi-agent tasks.
>
>   - **Progress-adaptive meta-plan and execution strategy** that dynamically adjusts and refines the meta-plan based on real-time progress, ensuring adaptability and responsiveness in unpredictable environments. This adaptiveness addresses the limitations of static or pre-defined strategies, particularly in scenarios with unexpected challenges.
>
> - These contributions advance the field of LLM-driven embodied agent cooperation by introducing a **dynamic escalation mechanism**. Unlike traditional methods like CoELA, this mechanism enables agents to actively detect and discuss potential errors or inconsistencies, particularly in decentralized settings with partial observations. Such environments often lack shared global information, leading to low cooperation efficiency, as indicated in Table 3. By introducing active intervention, our strategies prevent error propagation and significantly enhance system robustness and efficiency, as demonstrated in Tables 1 and 2.
>
> - Our strategies are indeed implemented through LLM prompting—a standard technique in LLM-based embodied agent methods like CoELA, RoCo [Mandi et al., 2023], and ProAgent [zhang2023proagent], and we never claimed otherwise. We emphasize that our novelty lies in the **design and mechanisms** of these strategies (meta-plan generation and its progressive adaptation) and the notable impact on agent cooperation, rather than the use of prompting itself, which is simply the ‘tool’.

---

> > ### Author Response · Authors · 2024-11-20
> > **Response To Reviewer 8MR1**
> >
> > ***Q3. The author should add notation of cited results for  RHP\*, RHP, and LLAMA-2 CoELA. Why the results  in Table 2 and Appendix Table 5 are organized as such needs further clarification.***
> >
> > - For RHP\*, RHP, and CoELA in Table 1, per your suggestion, we updated the manuscript to explicitly clarify that their results are quoted from the vanilla CoELA paper.
> >
> > - In the main manuscript, we comprehensively report the performance of all baselines, including CoELA, RoCo, and ProAgent, under GPT-3.5 and LLAMA 2 settings. These results clearly demonstrate the superiority of our CaPo framework across these two settings
> >
> > - In the Appendix Table 5, we further compare our method only with CoELA under the GPT-4 setting due to two key reasons:
> >   - The cost of the GPT-4 API is significantly higher—approximately 5 times that of GPT-3.5—which exceeded our limited budget, preventing us from testing RoCo and ProAgent under this setting.
> >   - Space constraints in the main manuscript prevent the inclusion of additional experiments under the GPT-4 setting, as Table 1 is already fully utilized.
> >
> > - Additionally, we compare CaPo with CoELA, RoCo, and ProAgent on the C-WAH task under GPT-4 in Table 3. These results further confirm the superior performance of CaPo over these baselines. Given the limitations in budget, page length, and our experimental focus, we believe the selected experimental reports are both sufficient and well-justified.
> >
> > - To further address your concerns, we also conducted additional experiments by evaluating RoCo and ProAgent on the TDW-MAT task under the GPT-4 setting. The results, provided below, demonstrate that CaPo consistently outperforms these baselines under GPT-4 setting. These supplementary results have been included in the Appendix, and will be incorporated into the final version of the manuscript, since we require additional time to carefully reorganize the content due to the page limitations.
> >
> > |           | RHP* | RHP | CoELA | Pro.Agent | RoCo | CaPo (ours) |
> > |:---------:|:----:|:---:|:-----:|:---------:|:----:|:-----------:|
> > | **Food**  |  52  |  76 |   87  |    84 (-3.4%)  |  88 (+1.1%) |   90 (+3.4%)   |
> > | **Stuff** |  49  |  74 |   83  |    85 (+2.4%)  |  82 (+1.2%) |   87 (+4.8%)   |
> > | **Avg.**  |  50  |  75 |   85  |    84 (-1.2%)  |  85 (+0.0%) |   89 (+4.7%)   |
> >
> > ***Q4.It is not specifically discussed whether this costly communication is still employed, and if so, some analysis around the communication cost and corresponding efficiency achieved may provide valuable insights for the community.***
> >
> > - We see communication cost and efficiency as important aspects that we may address from a different perspective. Regarding the cost (consuming time), it mainly includes the cost of the agent moving and the communication cost. Between the two,  the moving cost is often much higher than communication cost, since current robots lack the ability of humans, and their movement is costly especially in complex and large-scale environments like urban areas; while the latter only needs LLM’s inference, and massage sending and receiving, and is indeed much easier and efficient. For example, moving from one room to another can often require much more time cost than LLM’s inference.
> >
> > - To reduce communication cost, we adopt two strategies in CaPo: a) the agent’s progress-adaptive planning module evaluates whether the meta plan is suitable for the current task progress: if yes, then all agents will continue their subtasks, and will not communicate, reducing communication costs; b) with the meta plan as a reference, a few communication rounds are sufficient to arrive at a new plan. Indeed, we set the maximum round of communication as 3 for each meta-plan discussion. So these strategies differ from CoELA which needs communication at each iteration, while ours does not.
> >
> > - To further address your concerns, we report the total running time on the CWAH dataset, including moving and communication time. For visual Object setting on the CWAH task,  CoLEA and our CaPo respectively complete all tasks in 56.7 and 62.5 minutes, while needing an average of 92 and 83 moving steps on each task. This shows CaPo’s balanced runtime efficiency without significant added delays.
> >
> > |           | Running time cost on 10 tasks | Average steps on 10 tasks |
> > |:---------:|:-----------------------------:|:--------------------------:|
> > | **CoELA** |        56.7 min               |            92              |
> > | **CaPo (ours)** |    62.5 min               |            83              |

---

> > > ### Author Response · Authors · 2024-11-20
> > > **Response to Reviewer 8MR1**
> > >
> > > ***Q5. Is the progress-adaptive planning module always triggered whenever there is new progress? How many rounds of meta-plan updates are there on average and how many steps would be used to conduct this meta-plan discussion? Is the communication module also needed after the progress-adaptive planning module makes the update to share it with the others?***
> > >
> > > - **The progress-adaptive planning module is only triggered and activated when significant progress is made**, such as discovering new objects or completing a subtask (as shown in Figure 4). This mechanism preserves the effectiveness of the meta-plan and avoids redundant steps caused by an outdated plan.
> > >
> > > - For average rounds of meta-plan updates and average steps for each meta plan discussion,  we report the statistics below. Regarding Food tasks on TDW, agents update the meta-plan for an average of 14 times per task, with each discussion requiring 2.4 steps and a total 1,757 characters for sending (i.e., 4 frames since 1 frame can send 500 characters as in CoELA). While this incurs some time cost, the long-term coherent meta-plan results in a substantial transport rate  gain of +18.5%. This demonstrates that creating a coherent meta-plan through communication is significantly more time-efficient than relying solely on embodied actions.
> > >
> > > | Method     | # Round of meta-plan updates | # Step / communication of each meta-plan update | Time cost per meta-plan update | Time cost per task | Character cost of each meta-plan update | Transport rate gain |
> > > |:----------:|:----------------------------:|:-----------------------------------------------:|:-------------------------------:|:------------------:|:---------------------------------------:|:------------------:|
> > > | **Food tasks** | 14 times                   | 2.4 rounds                                       | 20.4 seconds                    | 26.2 mins          | 1757 characters                         | +18.5%             |
> > > | **Stuff tasks** | 17 times                   | 2.1 rounds                                       | 22.7 seconds                    | 25.6 mins          | 1462 characters                         | +15.1%             |
> > >
> > > When the meta-plan is updated, the main agent needs to use a communication module for sharing the updated meta plan to all other agents. Otherwise, a communication module will not be needed.
> > >
> > > ***Q6. What happens if two agents make new progress at the same time and all want to be the meta-plan designer? What will happen if agents can't reach a consensus? Will they just have no communication afterward or try to communicate again after how many steps or stuck at arguing every few steps?***
> > >
> > > - **Regarding simultaneous progress**, for each task, we designate one agent as the fixed meta-plan designer, while the other acts as the meta-plan evaluator. When both agents make progress simultaneously, the meta-plan designer promptly updates the meta-plan based on its own progress and then communicates with the evaluator to collaboratively refine and finalize the plan. This process allows the evaluator to assess whether the updated meta-plan aligns with its current state. If the evaluator identifies issues, it provides feedback, including reasons such as its current progress, to facilitate a more informed discussion and ensure the meta-plan is well-suited for both agents.
> > >
> > > - As illustrated in Figure 8, agents typically reach a consensus on the updated meta-plan within three rounds of discussion in most cases (e.g., 78.9% of all "stuff" tasks). If consensus is not achieved within three rounds, the discussion concludes with the most recent meta-plan, which all agents adhere to until new progress prompts another discussion phase. This mechanism strikes a balance between ensuring meta-plan quality and minimizing communication overhead.

---

> > > > ### Comment · Reviewer_8MR1 · 2024-11-24
> > > > **Thanks for the response, but more concerns arise**
> > > >
> > > > Thank you for your detailed response and additional experiments. However, my concerns about the experiments and contribution have not been fully addressed.
> > > >
> > > > **Experiments**
> > > >
> > > > - It was reported in the original paper that the CoELA method needs a strong backbone like GPT-4 to work well. So it doesn't sound fair to claim a 16\% improvement over CoELA but with a weaker GPT-3.5 backbone. Understandably, there may be budget constraints on conducting experiments with GPT-4, but this impacts the reported results' soundness nevertheless. I'm glad to see the authors have evaluated all baselines using GPT-4 in the rebuttal, which surely strengthens the soundness of the work. It may be more sound to report GPT-4 results consistently in the main paper and incorporate the discussion of such results into the main results analysis. An improvement of 16\%  or 4\% is quite different. The authors need to reorganize the results section and clarify this.
> > > >
> > > > - It's somewhat confusing why the authors reported *the relative improvement when comparing with baseline CoELA* in the main tables rather than the *efficiency Improvement (EI) of cooperating with other agents* reported in the original benchmark. The latter seems a more valid metric since it reports the relative performance improvements when have cooperators. Comparing other baselines to a specific CoELA baseline doesn't convey much information on the task itself.
> > > >
> > > > - In the original paper, there were also experiments of heterogeneous agents' cooperation, it would be great to incorporate such results here as well, such as RHP+CaPo. This could also help address the concerns over the contribution mentioned below.
> > > >
> > > > - The running time includes both the model's inference time (run models locally or call APIs, dependent on the hardware or network connections) and the simulator's simulation and render speed (which is dependent on the number of steps), which does not seem a good metric to evaluate the efficiency of the method. From Appendix B, communication is defined as an action that costs steps dependent on the message length in the environment, so it may be better to report the steps used for communicating as the communication cost.
> > > >
> > > > - Does the steps reported in the main table include the steps used to communicate or not? From Appendix B, communication is defined as an action that costs steps dependent on the message length in the environment. From the original CoELA paper, their reported results include the communication steps (Figure 4(c) in the CoELA paper). But quoted from the authors' response to Q4, "needing an average of 92 and 83 moving steps on each task.", seems only steps used for moving are reported here. **This seems an unfair comparison** and needs further explanation from the authors.
> > > >
> > > > **Contribution**
> > > >
> > > > - My concern over the contribution here is not about the use of prompting but this **pre-defined communication protocol**. It's not surprising to see a performance gain after assuming the cooperators will design and discuss the meta-plan in a predefined format. I wonder how this pre-defined protocol may work when cooperating with another agent that does not have such an assumption, such as MHP or humans. On the other hand, other baselines do not assume the existence of such a common ground and could be potentially deployed to cooperate with any other decentralized agents.
> > > >
> > > > Given the significantly weakened soundness of the experiments and the contribution, I'm **temporarily** adjusting my score to 5 to reflect this.

---

> > > > > ### Author Response · Authors · 2024-11-25
> > > > > **Second Round Response to 8MR1**
> > > > >
> > > > > Thank you for your thorough review and constructive suggestions. We have carefully considered your comments and have provided detailed responses below, hoping to address your concerns.
> > > > >
> > > > > ***Q1: Reorganization of main results and report GPT-4 results consistently in the main paper.***
> > > > >
> > > > > Thank you for your valuable suggestion regarding the reorganization of the main results. In response, we have revised Table 1 to ensure a fair and consistent comparison across all baselines, including CoELA.
> > > > >
> > > > > Specifically,  we now **present results using GPT-4 for CoELA, ProAgent, RoCo and our CaPo as the main results in Table1**.  Additionally, we also include the results for CoELA and our CaPo with other LLMs, i.e., GPT-3.5 and LLAMA-2, as additional results in Table 1. Finally, **we add some discussion and clarification of these results in LINE425-31**.
> > > > >
> > > > > We hope this reorganization and detailed reporting could address your concerns and provide a more comprehensive comparison with all baseline models.
> > > > >
> > > > > ***Q2. Reporting Efficiency Improvement (EI) Instead of Relative Improvement***
> > > > >
> > > > > Thank you for highlighting this point. We agree that Efficiency Improvement (EI) is a more valid and meaningful metric for performance comparison.
> > > > >
> > > > > In response to your suggestion, **we have revised Table 1 and Table 2 to report the main efficiency results with EI** rather than relative improvement over the baseline CoELA.
> > > > >
> > > > > We hope that this revision ensures a more consistent and fair comparison across all baseline models, and addresses your concern regarding the evaluation metrics.
> > > > >
> > > > > ***Q3.  Contribution concerns about pre-defined communication protocol and experiments on heterogeneous agents' cooperation.***
> > > > >
> > > > > We agree that our CaPo introduces a predefined communication protocol, and it is very interesting to explore how CaPo may work when cooperating with other agents like RHP, CoELA or human.   To address your concerns, **we conduct the experiments on heterogeneous agents, where CaPo cooperates with RHP and CoELA**.  We conduct experiments on the TDW-MAT tasks using oracle perception and GPT-4, and the results are shown below.
> > > > >
> > > > > | Methods           | Food (↑) | Stuff (↑) | Avg. (↑) |
> > > > > |----------------|----------|-----------|----------|
> > > > > | **RHP†**          | 0.52     | 0.49      | 0.50     |
> > > > > | **RHP+RHP†**      | 0.76 (+33%) | 0.74 (+34%) | 0.75 (+33%) |
> > > > > | **RHP+CoELA†**     | 0.85 (+40%) | 0.77 (+35%) | 0.81 (+37%) |
> > > > > | **RHP+CaPo**       | 0.85 (+38%) | 0.80 (+37%) | 0.82 (+37%) |
> > > > > | **CaPo+CoELA**     | 0.86 (+36%) | 0.83 (+35%) | 0.84 (+35%) |
> > > > >
> > > > > - As shown in the above table, when **CaPo cooperates with RHP agents**, it achieves a higher transport rate compared to both a team of two MHP agents and a combination of MHP and CoELA agents. We attribute this improvement to the independently developed meta-plan, which guides the CaPo agent to complete tasks more efficiently and implicitly enhances multi-agent cooperation efficiency.
> > > > >
> > > > > - Interestingly, when our **CaPo cooperates with the more advanced agent CoELA**, it further improves the transport rate. This highlights the potential of deploying CaPo to collaborate with other decentralized agents.
> > > > >
> > > > > - We have included these results in Appendix A2 of the Supplementary Materials. Additionally, **we have added the results of CaPo cooperating with MHP on the C-WAH tasks in Table 2**, along with discussions about these findings in **LINES 458-460**.
> > > > >
> > > > > We hope these additions provide a more comprehensive understanding of CaPo's collaborative capabilities and address your insightful comments.

---

> ### Author Response · Authors · 2024-11-25
> **Second Round Response to Reviewer 8MR1**
>
> ***Q4. It may be better to report the steps used for communicating as the communication cost.***
>
> Thank you for this valuable suggestion. In our CaPo framework, **communication is considered costly, with 500 characters equating to a cost of 1 frame**. Therefore, we agree that **reporting the number of steps (frames) used for communication** offers a more accurate representation of the communication cost.
>
> Accordingly, we provide a detailed analysis of the communication cost in the following table by collecting statistics on the TDW-MAT tasks.
>
> | Task     | Rounds of Meta-plan Updates | Character Cost per Meta-plan Updates | Frames Cost from Communication | Rounds of Discussion per Meta-plan Updates | Time Cost per Meta-plan Update | Time Cost per Task |
> |-------------|-------------------------------|------------------------------|--------------------------------|---------------------------------------|-------------------------------|--------------------|
> | **Food**  | 14                            | 1,757 chars                  | 56 frames                      | 2.4 rounds                           | 20.4 sec                      | 26.2 min          |
> | **Stuff** | 17                            | 1,462 chars                  | 51 frames                      | 2.1 rounds                           | 22.7 sec                      | 25.6 min          |
>
> - Taking the food tasks as an example, our CaPo requires 14 meta-plan updates per task, with each update generating an average of 1,757 characters. Since 500 characters equate to 1 frame, this results in **a communication cost of 56 frames out of the total 3,000 frames for each task**. Similarly,  the communication cost for each task on Stuff is 51 frames.
>
> - Additionally, all steps (frames) consist of both moving steps and communication steps. Therefore, **the communication cost is inherently reflected in the final efficiency metric**, i.e., the transport rate and average steps. As shown in the table 1 and 2, our method exhibits significant efficiency improvement, highlighting a good tradeoff between communication cost and efficiency  improvement. We include these results and analysis in A.6 of supplementary materials.
>
> We hope such analysis could address your concern about communication cost.
>
> **Q5. Does the steps reported in the main table include the steps used to communicate or not? This seems an unfair comparison.**
>
> We appreciate your thoughtful question. The steps reported in the main table **include both the steps used for communication and those used for movement**. Regarding our previous response to Q4, we apologize for the typo in stating "needing an average of 92 and 83 moving steps on each task." To clarify, **92 and 83 refer to the total number of steps, which include both communication and movement**.

---

> > ### Comment · Reviewer_8MR1 · 2024-11-26
> > **Thanks for the response, but still some issues**
> >
> > I appreciate the author's effort in re-organizing the results, it looks much better to me now. But there are still some issues I'd like to point out.
> >
> > - It seems there are still some typos in the response, e.g. in Q3, is it RHP or MHP the authors are talking about here? "when CaPo cooperates with RHP agents, it achieves a higher transport rate compared to both a team of two MHP agents and a combination of MHP and CoELA agents."
> >
> > - The numbers with EI reported seem confusing to me. Take the third and fourth rows in the Q3 Table as an example, RHP+CoELA and RHP+CaPo have the same transport rate of 0.85, but their EI is different. In my understanding, in the same column, the EI should be the same if the transport rate is the same.
> >
> > - The reported results show CaPo+CoELA performs slightly worse than CoELA+ CoELA (84 v.s. 85), which seems to verify my concern over the effectiveness of this pre-defined communication protocol when collaborating with agents that do not share such assumptions.
> >
> > - I encourage the authors to take time to carefully review their manuscripts esp. the experiments and make the contributions and limitations clearer.
> >
> > - The main paper seems to exceed the page limit now.

---

> ### Author Response · Authors · 2024-11-26
> **Third Round Response to Reviewer 8MR1**
>
> Thank you sincerely for your insightful comments, which greatly help us improve the quality of this manuscript. We provided detailed responses to the existing issues below.
>
> ***Q1. Typos in previous response to Q3.***
>
> Thank you for pointing out the potential typo in our response to Q3. We would like to clarify that the correct term should indeed be **RHP** in this context. Specifically, the sentence should read:
>
> *“When CaPo cooperates with RPH agents, it achieves a higher transport rate compared to both a team of two RHP agents and a combination of RHP and CoELA agents.”*
>
> We sincerely apologize for this oversight and have ensured all such typos are corrected in the revised manuscript. Thank you again for your attention to detail.
>
> ***Q2. Confusing EI numbers in the previous Q3 table.***
>
> Thank you for pointing out the discrepancy in EI values between RHP+CoELA and RHP+CaPo in the previous Q3 Table. We understand how this might seem confusing, and we clarify as follows:
>
> - **Source of Results**: The TR values for RHP and RHP+CoELA have been referenced directly from the CoELA paper, whereas the results for RHP+CaPo have been derived from our own experiments. Differences in result sources can lead to slight variations in the baseline performance of RHP, influenced by factors such as implementation details, hardware setups, or random seeds.
>
> - **Clarification with an Example**: For example, if the baseline TR for RHP in CoELA has been 0.51 and RHP+CoELA achieves a TR of 0.85, the EI for RHP+CoELA would be approximately 40%. In contrast, in our experiments, the baseline TR for RHP might have slightly differed, e.g., 0.52. While the TR for RHP+CaPo also reaches 0.85, the EI would be approximately 38.8%. This small variation in the baseline explains the difference in EI values, even when the final TR values are identical.
>
> - **Clarification in Manuscript**: To avoid any further confusion, we have explicitly indicated the sources of baseline results, Lines 801-2 of Appendix, and explained how these variations can affect EI calculations, even for identical TR values.
>
> We appreciate your careful attention to this detail and have ensured that these points are clearly addressed in the revised manuscript. Thank you for your valuable feedback.
>
> ***Q3. Concern about the pre-defined communication protocol.***
>
> Thank you for raising the concern about the pre-defined communication protocol, as reflected in the slightly worse performance of CaPo+CoELA compared to CoELA+CoELA (84 vs. 85). We provide the following clarification:
>
> - **Robustness of CaPo in heterogeneous settings**: The slight performance drop is predictable, as CaPo+CoELA involves heterogeneous agents, where CaPo independently generates and updates meta-plans. This requires CaPo to adapt its strategies to align with CoELA’s planning and communication mechanisms, which may not be fully compatible. Beside, RHP+CaPo also achieves slightly higher efficiency than RHP+CoELA (82 vs 81), demonstrating the feasibility of CaPo cooperating with diverse agents without pre-defined communication protocol.
>
> - **Higher Efficiency in Homogeneous Settings**: In homogeneous settings, CaPo+CaPo demonstrates significantly higher efficiency compared to CoELA+CoELA (89 vs 85). This improvement is attributed not only to the pre-defined communication protocol but also to our proposed meta-plan mechanism and progress-adaptive meta-plan updating. We hope these findings will inspire further advancements in the community.
>
> - **Acknowledging Limitations**: Slight worse performance compared with CoELA+CoELA highlights the inherent challenges of  CaPo in heterogeneous multi-agent cooperation setting . We have added a discussion of this limitation in LINE 536-39 to reflect this insight.
>
> We appreciate your thoughtful feedback and hope this explanation clarifies the observed results and addresses your concerns.

---

> > ### Author Response · Authors · 2024-11-26
> > **Third Round Response to Reviewer 8MR1**
> >
> > ***Q4. Authors should carefully review their manuscripts esp. the experiments and make the contributions and limitations clearer.***
> >
> > We sincerely appreciate your valuable suggestion. Following your advice, we have carefully reviewed the manuscript and made the following improvements:
> >
> > - **Experiments**: To enhance clarity and comprehensiveness, we have made the following updates:
> >    - Added a detailed introduction to the Efficiency Improvement (EI) metric in Lines 370–372 to improve clarity and readability.
> >    - Improved the organization of Tables 1 and 2, with clearer discussions and analyses in Lines 420–428 and Lines 431–457.
> >    - Included a description of the heterogeneous agents baseline in Lines 410–411.
> >    - Clarified the convergence analysis of discussion rounds and introduced the corresponding communication cost in Lines 515–524.
> >    - Added experimental results and analysis on heterogeneous agents’ cooperation in Appendix A2.
> >    - Included additional experiments and analysis on the RoCObench environment in Appendix A4.
> >    - Provided a detailed analysis of the number of discussion rounds in Appendix A5.
> >    - Elaborated on communication cost analysis in Appendix A6.
> >    - Introduced experiments on collaborating with humans in Appendix A7.
> >
> > - **Contributions**: To make our contributions clearer and more distinct:
> >
> >     - Improved the description of our contributions in the introduction, emphasizing key innovations such as the meta-plan mechanism and the progress-adaptive meta-plan and execution and how they address core challenges in multi-agent cooperation.
> >     - Reorganized the methods section by moving modules inherited from CoELA, such as perception, to the preliminary section. This refocuses the methods section on our proposed approach, making our contributions more prominent and distinct.
> >
> > - **Limitations**: Added an analysis of our method’s limitations in Lines 536–539. This discusses the reliance on LLMs for reasoning and planning, as well as the challenges posed by heterogeneous environments without predefined communication protocols.
> >
> > We hope these revisions effectively address your concerns and improve the clarity of the manuscript. Thank you once again for your insightful feedback and guidance.
> >
> > ***Q5. The main paper seems to exceed the page limit now.***
> >
> > Thank you for pointing this out. We have already addressed this issue and ensured that the main paper now adheres to the page limit.

---

> > > ### Author Response · Authors · 2024-11-29
> > > **Gentle Reminder**
> > >
> > > Dear Reviewer,
> > >
> > > The discussion period closes in few days. We've tried to address all your concerns with new results, clarifications and an updated manuscript. Please let us know if you have any remaining concerns. We look forward to a productive discussion.
> > >
> > > Best,
> > >
> > > Authors

---

> > > > ### Comment · Reviewer_8MR1 · 2024-11-30
> > > > **Thanks for the response**
> > > >
> > > > Thanks for the authors' continual responses. After making the contribution and experiments clearer, my major concerns were addressed. I'm raising my score back to 6, and I'm leaning toward the acceptance of this paper.

---

### Meta-Review · Area_Chair_5pmS · 2024-12-20

**Metareview:**

This paper proposes an LLM-based embodied cooperation method that extends prior methods such as CoELA with a meta plan generation and dynamic adjustment, inspired by human cooperation. This is a meaningful contribution to LLM-based multi-agent cooperation. The revision and additional results further strengthen the submission. Overall, the proposed method has demonstrated a significant improvement over existing methods on multiple challenging benchmarks.

**Additional Comments On Reviewer Discussion:**

One reviewer raised their rating to 8 after the rebuttal. During AC-reviewer discussion, one reviewer clarified their remaining concern: evaluation on RoCoBench and exploring methods to reduce communication. However, the revision already provided results on RoCoBenchm, showing good performance. The authors also have addressed the cost of communication during rebuttal. Therefore, the AC thinks that the revision can be rated above acceptance.

---

### Decision · Program_Chairs · 2025-01-22

Accept (Poster)